

# Revised fractional abundances and warm-season temperatures substantially improve brGDGT calibrations in lake sediments

Jonathan H. Raberg[1,2], David J. Harning[1,2], Sarah E. Crump[1,*], Greg de Wet[1,†], Aria Blumm[1,‡], Sebastian Kopf[1], Áslaug Geirsdóttir[2], Gifford H. Miller[1], Julio Sepúlveda[1]

1, Department of Geological Sciences and Institute of Arctic and Alpine Research, University of Colorado Boulder, Boulder, CO, 80309, USA
2, Faculty of Earth Sciences, University of Iceland, Reykjavík, Iceland

*Currently at the Ecology and Evolutionary Biology Department, University of California Santa Cruz, Santa Cruz, CA 95064
†Currently at the Department of Geosciences, Smith College, Northampton, MA 01063
‡Currently at the Department of Geosciences, University of Arizona, Tucson, AZ 85721

*Correspondence to:* Jonathan H. Raberg (jonathan.raberg@colorado.edu)

**Abstract.** Distributions of branched glycerol dialkyl glycerol tetraethers (brGDGTs) are frequently employed for reconstructing terrestrial paleotemperatures from lake sediment archives. Although brGDGTs are globally ubiquitous, the microbial producers of these membrane lipids remain unknown, precluding a full understanding of the ways in which environmental parameters control their production and distribution. Here, we advance this understanding in three ways. First, we present 43 new high-latitude lake sites characterized by low mean annual air temperatures (MATs) and high seasonality, filling an important gap in the global dataset. Second, we introduce a new approach for analyzing brGDGT data in which compound fractional abundances (FAs) are calculated within structural groups based on methylation number, methylation position, and cyclization number. Finally, we perform linear and nonlinear regressions of the resulting FAs against a suite of environmental parameters in a compiled global lake sediment dataset (n = 182). We find that our approach deconvolves temperature, conductivity, and pH trends in brGDGTs without increasing calibration errors from the standard approach. We also find that it reveals novel patterns in brGDGT distributions and provides a methodology for investigating the biological underpinnings of their structural diversity. Warm-season temperature indices outperformed MAT in our regressions, with Months Above Freezing yielding the highest-performing model (adjusted $R^2$ = 0.91, RMSE = 1.97°C, n = 182). The natural logarithm of conductivity had the second-strongest relationship to brGDGT distributions (adjusted $R^2$ = 0.83, RMSE = 0.66, n = 143), notably outperforming pH in our dataset (adjusted $R^2$ = 0.73, RMSE = 0.57, n = 154) and providing a potential new proxy for paleohydrology applications. We recommend these calibrations for use in lake sediments globally, including at high latitudes, and detail the advantages and disadvantages of each.

## 1 Introduction

Paleotemperature records from lake sediment archives are highly sought after in studies of terrestrial paleoclimate. Bacterial branched glycerol dialkyl glycerol tetraether (brGDGT) lipids have solidified themselves as an important tool in this pursuit (Fig A1; Schouten et al., 2013). First isolated from peat (Sinninghe Damsté et al.,



2000), these membrane lipids have since been measured in increasingly diverse settings, from marine, soil, lacustrine,
and riverine locations (Hopmans et al., 2004; Weijers et al., 2006; Pearson et al., 2011; De Jonge et al., 2014b,
respectively) to hot springs, fossil bones, groundwater, deep ocean trenches, and methane seeps (Li et al., 2014; Dillon
et al., 2018; Ding et al., 2018; Xiao et al., 2020; Zhang et al., 2020, respectively). Their ubiquity in nature has given
them widespread applicability as environmental proxies; brGDGTs have been used to reconstruct temperature in a
variety of archives including lake sediments (e.g. de Wet et al., 2016), marine sediments (e.g. Dearing Crampton-
Flood et al., 2018), peat (e.g. Zheng et al., 2017), loess (e.g. Lu et al., 2019), and fossil bone (e.g. Zhao et al., 2020).
They have additionally been used to reconstruct lake water pH (e.g. Cao et al., 2017). As the microbial producers of
brGDGTs remain elusive (Sinninghe Damsté et al., 2018), these paleoclimate reconstructions currently rely on
empirical calibrations at both the regional and global level.

At the heart of brGDGT calibrations is the observation that the degree of alkyl-chain methylation and

cyclization are correlated to environmental temperature and pH, respectively. These relationships were first quantified
by the Methylation and Cyclization of Branched Tetraether indices (MBT and CBT) in a global soil dataset (Weijers
et al., 2007). The authors proposed physiological explanations for both connections, positing that an increase in
methylation number will enhance membrane fluidity, a desirable trait in cold environments, while a greater number
of cyclic moieties could improve proton permeability, an advantageous adaptation at high pH. These physiological
responses have precedent in other bacterial lipid classes (Reizer et al., 1985; Beales, 2004; Yuk and Marshall, 2004)
and appear to function for brGDGTs as well. However, genomic analyses of environmental samples (Weber et al.,
2018; De Jonge et al., 2019; van Bree et al., 2020) have suggested that differences in brGDGT distributions may also
stem from shifts in bacterial community composition. Variations in the position of alkyl-chain methylations (Fig. A1;
De Jonge et al., 2014a) further complicate the picture, with most studies showing 5-methyl brGDGT isomers to
correlate better with temperature than their 6-methyl counterparts (Russell et al., 2018), but others arriving at the
opposite result (Dang et al., 2018). These isomeric variations have additionally been shown to correlate with pH in
lake sediments (Dang et al., 2016). These discoveries highlight the multifaceted nature of the empirical relationship
between brGDGTs and environmental gradients and the need for further study.

Without a clear mechanistic understanding of brGDGTs' dependencies on environmental parameters and no

brGDGT-producing model organisms currently available for laboratory experimentation, researchers have relied on
statistical methods to construct empirical brGDGT calibrations. The majority of recent calibrations have employed a
variety of statistical techniques to construct linear or polynomial regressions using brGDGT fractional abundances
(FAs; De Jonge et al., 2014a; Martínez-Sosa et al., 2020b; Pérez-Angel et al., 2020). The fractional abundance $fx_i$ of
a compound $x_i$ in a set of $n$ compounds is defined as,

$$fx_i = x_i/(x_1 + x_2 + \cdots + x_n) \tag{1}$$

where any $x$ is the absolute abundance of the given compound in the set. For brGDGTs, these FAs are traditionally
calculated using all 15 commonly-measured compounds (Fig. A1),

$$fx_i = x_i/(Ia + Ib + Ic + IIa + IIb + IIc + IIIa + IIIb + IIIc + IIa' + IIb' + IIc' + IIIa' + IIIb' + IIIc') \tag{2}$$

where $x_i$ is any given compound in the denominator. By grouping together all 15 common brGDGTs, this approach
makes no prior assumptions about the relationships between the compounds themselves and maximizes the degrees



of freedom available when exploring a dataset. As relationships between brGDGTs and environment parameters are
not yet fully understood, this indiscriminate approach is appropriate. However, by lumping compounds of various
types and abundances into the denominator, the approach can also dampen meaningful trends and obscure important
relationships, especially for less abundant molecules. This adverse effect has been recognized for other lipid
biomarkers and has led to, for example, the exclusion of crenarchaeol from the $TEX_{86}$ index (Schouten et al., 2002)
and tetra-unsaturated alkenones from the $U^{K'}_{37}$ index (Prahl and Wakeham, 1987). Numerous ratio-based indices have
been developed for brGDGTs that similarly exclude low-abundance (e.g. MBT'; Peterse et al., 2012) or problematic
(e.g. $MBT'_{5Me}$; De Jonge et al., 2014a) compounds. However, a selective approach to fractional abundance
calculations has been hitherto unexplored.
On the other side of the calibration equations are the environmental variables that are regressed against
brGDGT indices and FAs. Mean annual air temperature (MAT) has been the traditional target of brGDGT calibrations
in lake sediments (e.g. Tierney et al., 2010; Loomis et al., 2012). However, it was recognized early on that brGDGT-
derived temperatures in cold regions may more accurately reflect warm-season temperatures (Pearson et al., 2011;
Sun et al., 2011), an hypothesis that was strongly supported in high-latitude lake sediments (Shanahan et al., 2013;
Peterse et al., 2014; Foster et al., 2016). Since the methodological advances that allowed for the separation of 5- and
6-methyl isomers (De Jonge et al., 2014a) and the development of new calibrations, both modern (Hanna et al., 2016;
Dang et al., 2018; Cao et al., 2020) and paleo (Super et al., 2018; Thomas et al., 2018; Crump et al., 2019; Harning et
al., 2020) studies have continued to support a warm-season bias. Additionally, a recent Bayesian calibration found the
mean temperature of Months Above Freezing (MAF) to be the only mode to significantly correlate with brGDGT
distributions in a global lake sediment dataset (Martínez-Sosa et al., 2020). However, this warm-season bias has yet
to be tested thoroughly in the regions in which it is most pronounced – namely, those with low MAT and high
seasonality. As these are the regions that are currently experiencing the most rapid climate change (Landrum and
Holland, 2020), their temperature histories are of high interest (Miller et al., 2010) and the quantification of the
brGDGT warm-season bias is an important target of study.
Outside of temperature, pH is the most common focus of calibration studies (e.g. Russell et al., 2018).
However, numerous other variables including conductivity (Tierney et al., 2010; Shanahan et al., 2013), dissolved
oxygen (DO; Colcord et al., 2017; Weber et al., 2018; van Bree et al., 2020; Yao et al., 2020), nutrient availability
(Loomis et al., 2014a), and lake mixing regime (Loomis et al., 2014b; van Bree et al., 2020) have been shown to be
potentially important controls on brGDGT distributions. Modern calibrations do not currently exist for these
environmental variables, largely due to the complexity of the relationships and data limitations.
In this study, we aim to improve lake sediment calibrations for brGDGTs in three ways. First, we extend the
global calibration dataset to include high-latitude sites by adding surface sediment from 43 lakes in the Eastern
Canadian Arctic, Northern Quebec, and Iceland. Second, we selectively group brGDGTs based on methylation
number, methylation position, and cyclization number, and use FAs calculated within these structural sets to
deconvolve environmental influences and identify novel patterns in brGDGT distributions. Finally, we analyze the
relationship between the compiled global dataset and MAT, four warm-season temperature indices, pH, conductivity,
DO, and lake geometry and generate empirical calibrations for use in lake sediments globally.




## 2 Methods

### 2.1 Study sites and sample collection

Surface sediments (0-0.5, 0-1, or 0-2 cm; Ekman box corer or core-top sediment) were collected from 43
lakes (26 from Iceland; 16 from Baffin Island, Arctic Canada; 1 from Northern Quebec; Fig. 1 insets) between 2003
and 2019. For 28 of these lakes, water temperature, pH, conductivity, and DO were measured at the time of sampling
during the summers of 2017-2020 using a multiparameter probe (HydroLab HL4, OTT HydroMet). These parameters
were additionally measured beneath the lake ice in Feb-May of 2018-2020 for 11 lakes. All but one of these lakes
experienced depleted bottom water oxygen levels under the ice relative to ice-free conditions. We therefore assume
that our ice-free water chemistry profiles do not capture the minimum DO ($DO_{min}$) levels experienced by our Canadian
and Icelandic lakes and exclude them from our analysis of $DO_{min}$, with the exception of the site from Northern Quebec,
which contained a summer oxycline. All other water chemistry parameters were averaged across all depths and seasons
before being used in calibrations.

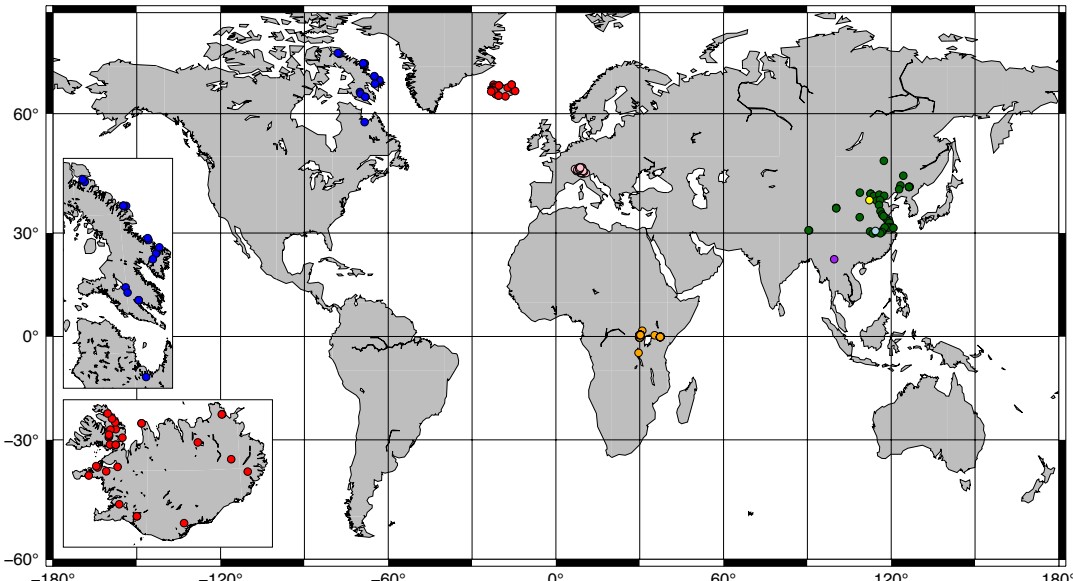


**Figure 1. Map of sites included in this study. Red and blue: Iceland and Canada (this study; see insets); Orange: Russell et al. (2018); Green: Dang et al. (2018); Pink: Weber et al. (2018); Yellow: Cao et al. (2020); Light blue: Qian et al. (2019); Purple: Ning et al. (2019).**

Previously published data from 36 lakes in Central Europe (Weber et al., 2018), 65 lakes in East Africa
(Russell et al., 2018), and 38 lakes in China (Dang et al., 2018; Ning et al., 2019; Qian et al., 2019; Cao et al., 2020),
were added to the dataset for a total of 182 data points (Fig. 1). Average brGDGT FAs were used for lakes with
multiple surface sediment samples (up to three in some cases). Lake surface areas (SAs) were taken from published
datasets or estimated using DigitalGlobe imagery. Lake volumes were estimated by approximating each lake basin as



a hemiellipsoid ($volume = 4/3 \times SA \times maximum\ depth$). Water chemistry parameters were taken from the
literature where available or else excluded from our analyses.

### 2.2 Sample extraction and analysis

Roughly 1 g of freeze-dried sediment was extracted using either an accelerated solvent extractor (ASE 200
DIONEX; 10 samples) or a modified Bligh and Dyer (BD; 33 samples) method. We provide a comparison of both
extraction methods below. For the ASE method, samples were extracted twice using 9:1 (v:v)
dichloromethane:methanol (DCM:MeOH) at 100°C and 2,000 psi. Total lipid extracts (TLEs) were redissolved in
99:1 (v:v) hexane:isopropanol (Hex:IPA) and filtered (0.45 μm, PTFE) before analysis. The remaining 33 samples
were extracted using a modified BD procedure (Wörmer et al., 2013). Briefly, sediment was vortexed and sonicated
in Mix A (DCM:MeOH:50mM Phosphate buffer (aq., pH 7.4) [1:2:0.8, v:v:v]). The mixture was then centrifuged at
3,000 rpm and 10°C for 10 minutes and the supernatant was collected in a glass separatory funnel. The process was
performed twice with Mix A, twice with Mix B (DCM:MeOH:5% Trichloroacetic acid buffer (aq., pH 2) [1:2:0.8,
v:v:v]), and once with Mix C (DCM:MeOH [1:5, v:v]). Equal volumes of HPLC-grade water and DCM were added
to induce separation. The organic fraction was collected and dried under a nitrogen stream. The aqueous phase was
washed once with DCM and the organic fraction was added to the extract. The TLE was then redissolved in 99:1 (v:v)
Hex:IPA and filtered (0.45-μm, PTFE) before analysis.
We analyzed brGDGTs using a Thermo Scientific UltiMate 3000 high-performance liquid chromatography
instrument coupled to a Q Exactive Focus Orbitrap-Quadrupole high-resolution mass spectrometer (HPLC-MS) via
an atmospheric pressure chemical ionization (APCI). We achieved chromatographic separation using a slightly
modified version (Crump et al., 2019; Harning et al., 2019; Pérez-Angel et al., 2020) of the HPLC method described
by Hopmans et al. (2016). Due to observed deterioration of chromatography over time, we lowered the initial
concentration of eluent B from 18% to 14% to maintain optimal separation of the 5- and 6-methyl isomers. A C46
GDGT internal standard (Huguet et al., 2006) was added to the TLE immediately after extraction and was used to
quantify brGDGT yields.

### 2.3 Comparison of ASE and BD Extraction Methods

To ensure that brGDGT distributions were agnostic to our extraction method, we extracted three surface
sediments, two suspended particulate matter (SPM) samples (2.5 L lake water filtered onto 0.3 μm glass fiber filters),
and three soils from Baffin Island using the ASE and BD methods in parallel (Fig. S1). The mean difference in
MBT'$_{5Me}$ (Eq. A3) between the two extraction methods was $0.006 \pm 0.004$, or $2 \pm 2\%$. This translates to a MBT'$_{5Me}$-
derived temperature difference of $0.2 \pm 0.2$°C using recent calibrations for soils (Naafs et al., 2017) and lake sediments
(Russell et al., 2018), which is well below their respective RMSEs of 5.3°C and 2.14°C (Fig. S2). To test for
compound-specific differences, we calculated percent differences in FAs between the two methods. For compounds
with FAs > 0.05 in the ASE method, the mean absolute percent difference compared to BD was $4 \pm 3\%$. For the lower
abundance compounds (FA ≤ 0.05), this difference was higher ($19 \pm 17\%$). No biases in the FA differences were
found in either case (difference < standard deviation).



We further extracted the BD sample residue with the ASE method to determine if any brGDGTs remained
after BD extraction. On average, we recovered only an additional $0.8 \pm 0.6\%$ brGDGTs. These residual brGDGTs had
a similar MBT'$_{5Me}$ to that of original BD extract (mean difference = $0.02 \pm 0.01$, equivalent to $0.5 \pm 0.4°C$) and were
not present in high enough abundances to significantly affect the overall BD distributions (Fig. S2). We therefore
conclude that there are no significant differences between samples extracted with the two methods and treat them
identically in the analyses that follow.

**2.4 Air Temperatures**
Monthly air temperature averages were gathered using the following methods. For nine sites on Baffin Island,
one year of *in situ* two-meter air temperature data from five temperature loggers (one- to four-hour resolution,
Thermochron iButtons, Maxim Integrated Products) was converted to a 30-year monthly climate normal (1971 to
2000) using a transfer function to relate local data to nearby meteorological stations (Department of Environment,
Government of Canada). For the remaining Canadian sites as well as all sites in Iceland, we used the WorldClim
database (Fick and Hijmans, 2017) to generate 30-year climate normals for the same time period (1970 to 2000).
Monthly temperatures for Central European sites were derived using monthly altitudinal lapse rates constructed from
climate normals (1970 to 2013) of 148 meteorological stations (Federal Office of Meteorology and Climatology:
MeteoSwiss). Monthly temperature data was not available for the East African lakes. However, the seasonality
(standard deviation of monthly temperatures) of these lakes is low ($0.5 \pm 0.2$ °C in the WorldClim database, or < 2%
of range of the dataset). We therefore approximate all monthly temperatures to be equivalent to MAT for these lakes.
Monthly temperature data from all other studies were either published or provided by the authors.
We used the above monthly air temperatures to calculate MAT and four warm-season temperature indices.
Three of these indices represent an average temperature for the warmer portion of the year: mean temperature of
Months Above Freezing (MAF), Mean Summer Temperature (MST; mean of June, July, and August in the Northern
Hemisphere and December, January, and February in the Southern Hemisphere), and mean Warmest Month
Temperature (WMT). These indices capture average temperatures for the warm season, but are unaffected by its
duration. We therefore additionally calculate the Summer Warm Index (SWI), defined as the cumulative sum of all
monthly temperatures above 0°C. This index represents an important control on vegetation patterns at high latitudes
and is a useful alternative to the Growing Degree Days Above 0°C (GDD$_0$) index when daily temperature data is not
available (e.g. Raynolds et al., 2008). For our five *in situ* temperature loggers in the Eastern Canadian Arctic for which
sub-daily temperature data is available, GDD$_0$ and SWI are highly correlated ($R^2 = 0.998$).

**2.5 Statistical Methods**
To construct calibrations between brGDGT FAs and environmental variables, we used the following method.
Each FA was first regressed alone against the environmental variable being investigated. Compounds with a
correlation p-value $\geq 0.01$ were considered non-significant and removed from further analysis (Pérez-Angel et al.,
2020). Fits were then constructed from the remaining compounds using two independent approaches. The first
approach was stepwise forward selection/backwards elimination (SFS/SBE) using the MASS package (Venables and



Ripley, 2002) in R (R Core Team, 2018). This approach finds the best fit by sequentially adding (SFS) or removing
(SBE) terms in a generalized linear model and evaluating the resulting fit using the Bayesian Information Criterion
(BIC; Schwarz, 1978). The approach is common for constructing brGDGT calibrations (e.g. Dang et al., 2018; Russell
et al., 2018), but it is not exhaustive. We therefore additionally used the leaps package (Lumley, 2020) in R to evaluate
all possible linear combinations of fitting variables (the "combinatoric" approach, Pérez-Angel et al., 2020). We again
used the BIC to select the best fit. For both methods, we imposed the additional criterion that each of the resulting
fitting variables must itself be statistically significant ($p < 0.01$). To help avoid overfitting, we additionally used the
adjusted $R^2$ to evaluate calibration performances. Some of our variables (conductivity, depth, surface area to depth,
and volume) spanned multiple orders of magnitude. For these variables, we performed regressions against the natural
logarithm of the variable.
We applied this calibration procedure to the FAs of each brGDGT structural set and subset defined below
(Sect. 3). In the subset-specific calibrations, it was sometimes possible for a single compound to dominate (FA = 1).
These samples were generally clear outliers resulting from the low natural abundances of all other members of the
subset and they were therefore removed from the subset-specific calibration models. We additionally tested for linear
regressions against a number of previously-defined brGDGT indices. A summary of these indices and their definitions
is provided in the Appendix.
All correlations reported in the text and figures were significant ($p < 0.01$) except those marked with an
asterisk or with a p-value provided. All $R^2$ values reported in the text and figures are adjusted $R^2$.

**3 Partitioning brGDGTs into structural sets for FA calculations**
The structure of brGDGTs can vary in three (currently-observed) ways: methylation number, methylation
position, and cyclization number. These variations result in 15 commonly-measured brGDGTs (Fig. A1). The standard
approach in brGDGT analysis is to calculate fractional abundances using all 15 of these compounds (Eq. 2). By
mathematically mixing brGDGTs with varying methylation number, methylation position, and cyclization number,
however, this approach risks convoluting the influences of disparate environmental variables. Here, we present a
method of grouping brGDGTs that highlights one type of structural variation (e.g. methylation number) while holding
one or both of the others constant. By calculating FAs within each group, we aim to deconvolve the influences of
temperature, pH, and other environmental variables on the structural variations of brGDGTs.
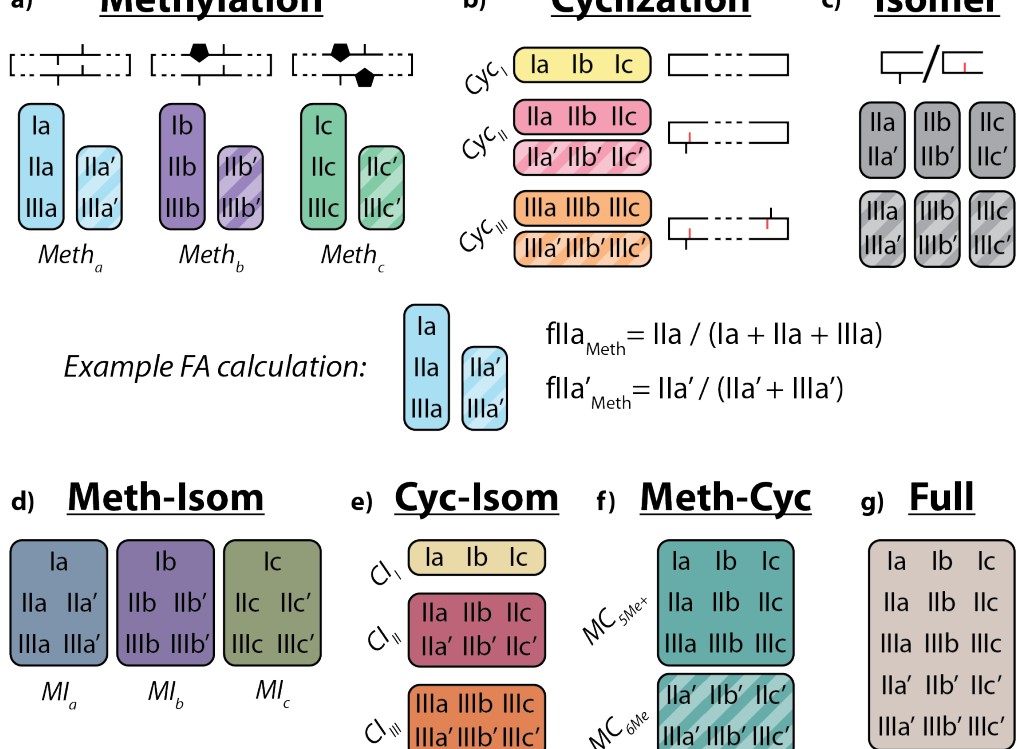

Figure 2. Schematic of the basic (a-c) and combined (d-g) brGDGT structural sets. Fractional abundances are calculated within each boxed group independently (Eq. (3-9) and Table A1). Schematic structures highlight the defining alkyl-chain moieties, with cyclopentane rings filled in for emphasis and C6 methylations denoted in red. Complete structures are available in Fig. A1.

To explore how changes in methylation number alone relate to environmental parameters, we constructed the Methylation (Meth) set (Fig. 2a). This set was generated by grouping brGDGTs with the same number of cyclopentane rings and the same methylation positions. Fractional abundances calculated within the Meth set solely reflect changes in methylation number, while ring number and isomer designation are held constant. Within the Meth set, we defined the Meth-5Me and Meth-6Me subsets as those that contained only 5- and 6-methyl lipids, respectively. As the tetramethylated brGDGTs (Ia, Ib, and Ic) are neither 5-methyl nor 6-methyl isomers, we generated versions of these two subsets that excluded (Meth-5Me, Meth-6Me) and included (Meth-5Me+, Meth-6Me+) these compounds (Fig. S3a-d). We found that the correlation between 5-methyl isomers and temperature was significantly improved when the tetramethylated compounds were included in their FA calculations, while the opposite was true for the 6-methyl compounds (see discussion in Sect. 4.2.2). We therefore grouped the tetramethylated compounds with the 5-methyl compounds (and not with the 6-methyl compounds) when generating the Meth set. FAs for the Meth set were calculated using Eq. (3),





$$fxy_{Meth} = xy \bigg/ \sum_{n=I}^{III} ny \; ; \; fxy'_{Meth} = xy' \bigg/ \sum_{n=II}^{III} ny' \qquad (3)$$

where $fxy$ and $xy$ are the fractional and absolute abundances of the brGDGT with Roman numeral $x$ (I, II, or III) and
alphabet letter $y$ (a, b, or c) and tetramethylated compounds are grouped with 5-methyl isomers.

We next defined the Cyclization (Cyc) set to examine the relationship of brGDGT ring number with

environmental variables (Fig. 2b). The Cyc set was formed by grouping brGDGTs with the same number and position
of methylations. With these variables held constant, variations in the Cyc FAs reflect only variations in the number of
cyclopentane moieties. We defined the Cyc-5Me and Cyc-6Me subsets as those containing only 5- and 6-methyl
isomers, respectively (Fig. S3e-f). FAs for the Cyc set were calculated using Eq. (4),

$$fxy^{(')}_{Cyc} = xy^{(')} \bigg/ \sum_{m=a}^{c} xm^{(')} \qquad (4)$$

where $fxy$ and $xy$ are the fractional and absolute abundances of the 5- or 6-methyl brGDGT with Roman numeral $x$ (I,
II, or III) and alphabet letter $y$ (a, b, or c).

Third, we defined the Isomer (Isom) set to isolate changes in the relative abundances of brGDGT isomers

(Fig. 2c). The Isom set was constructed by grouping brGDGTs with the same number of methylations and cyclizations.
Its FAs are solely a measure of the relative abundances of 5- and 6-methyl isomers, without the convoluting influence
of ring or methylation number variations. The isomeric diversity of brGDGTs is large, however, and there are
structural variations that are not controlled for within this set. For example, hexamethylated brGDGTs have two
isomers: one with methylations on different alkyl chains (e.g. at C5 and C5') and another with methylations on the
same chain (e.g. C5 and C24; De Jonge et al., 2013). As these compounds coelute, they cannot be treated independently
at this time. Additionally, it is unclear whether brGDGT-IIIa" (Weber et al., 2015), which contains both a C5 and a
C6' methylation, should be grouped with 5- or 6-methyl brGDGTs, or whether the three recently-identified 7-methyl
brGDGTs (Ding et al., 2016) should be included in the Isom series. As these compounds are rarely reported, we
excluded them from our analysis here, but suggest the possibility of expanding the Isom set to include them in the
future should more data become available. The Isom FAs were calculated using Eq. (5),

$$fxy^{(')}_{Isom} = xy^{(')} \bigg/ \sum_{isomers} xy \qquad (5)$$

where f$xy$ and $xy$ are the fractional and absolute abundances of the 5- or 6-methyl brGDGT with Roman numeral $x$ (I,
II, or III) and alphabet letter $y$ (a, b, or c) and "isomers" refers to 5- and 6-methyl brGDGTs. The Isom set contained
groups of two compounds each, making their FAs redundant. We therefore used only the 6-methyl FAs in our analysis.

The Meth, Cyc, and Isom sets each allow only one structural component of brGDGTs to vary. It is possible,

however, that two structural alterations occur in tandem in response to the same environmental variable. We therefore
defined three additional sets that hold one variable constant while allowing the other two to vary. The first is the Meth-
Isom (MI) combination set (Fig. 2d). In this set, brGDGTs with the same ring number are grouped together, while
both methylation number and position are allowed to vary. The Cyc-Isom (CI; Fig. 2e) is analogously constructed by
holding methylation number constant, while the Meth-Cyc (MC; Fig. 2f) set holds methylation position constant
(again treating tetramethylated brGDGTs as 5-methyl compounds, Fig. S3g-j). Finally, we defined the Full set (Fig.
2g), which takes the standard approach of allowing all three structural characteristics to vary freely by grouping all 15
commonly-measured brGDGTs together. The FAs of the combined sets are calculated using Eq. (6-9),

$$f\,xy^{(\prime)}{}_{MI} = xy^{(\prime)} \bigg/ \sum_{isomers} \sum_{n=I}^{III} ny \tag{6}$$

$$f\,xy^{(\prime)}{}_{CI} = xy^{(\prime)} \bigg/ \sum_{isomers} \sum_{m=a}^{c} xm \tag{7}$$

$$f\,xy_{MC} = xy \bigg/ \sum_{n=I}^{III}\sum_{m=a}^{c} nm\,; \quad f\,xy'_{MC} = xy' \bigg/ \sum_{n=II}^{III}\sum_{m=a}^{c} nm' \tag{8}$$

$$f\,xy^{(\prime)}{}_{Full} = xy^{(\prime)} \bigg/ \sum_{isomers} \sum_{n=I}^{III}\sum_{m=a}^{c} nm \tag{9}$$

where $fxy$ and $xy$ are the fractional and absolute abundances of the 5- or 6-methyl brGDGT with Roman numeral $x$ (I,
II, or III) and alphabet letter $y$ (a, b, or c) and tetramethylated compounds are treated as 5-methyl isomers. An expanded
guide to FA calculations is provided in Table A1.

As a proof of concept, we show that the fractional abundance of just one compound, brGDGT-Ia, can be

calculated within different structural sets to provide either a strong temperature or pH correlation, without a strong
cross-correlation. When the standard FA is calculated using all 15 compounds ($fIa_{Full}$), a moderate correlation is found
with MAF ($R^2 = 0.61$) and none with pH ($R^2 = 0.04$, p = 0.014; Fig. 3). When cyclization is held constant by using
the MI set, correlation with MAF increases ($R^2 = 0.75$) while that with pH remains uncorrelated ($R^2 = 0.13$). The
temperature correlation increases further when isomer designation is controlled for as well ($fIa_{Meth-5Me+}$, $R^2 = 0.88$),
with an $R^2$ nearly matching that of MBT'$_{5Me}$ ($R^2 = 0.89$) and the pH correlation remaining low ($R^2 = 0.27$). In contrast,
the correlation with temperature disappears for the analogous 6-methyl subset ($fIa_{Meth-6Me+}$, $R^2 = 0.08$). Finally,
allowing only ring number to vary ($fIa_{Cyc}$) effectively erases the correlation with temperature ($R^2 = 0.18$) and instead
provides a correlation with pH that, while modest, is already higher than any reported for a lake sediment calibration
to date ($R^2 = 0.59$).

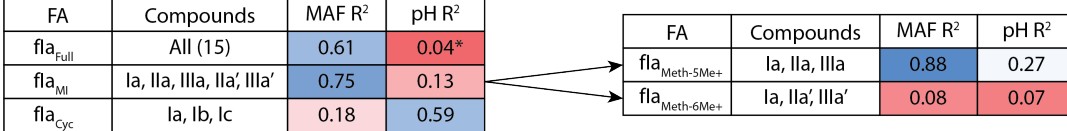

| FA | Compounds | MAF $R^2$ | pH $R^2$ |
|---|---|---|---|
| $fIa_{Full}$ | All (15) | 0.61 | 0.04* |
| $fIa_{MI}$ | Ia, IIa, IIIa, IIa', IIIa' | 0.75 | 0.13 |
| $fIa_{Cyc}$ | Ia, Ib, Ic | 0.18 | 0.59 |

| FA | Compounds | MAF $R^2$ | pH $R^2$ |
|---|---|---|---|
| $fIa_{Meth-5Me+}$ | Ia, IIa, IIIa | 0.88 | 0.27 |
| $fIa_{Meth-6Me+}$ | Ia, IIa', IIIa' | 0.08 | 0.07 |

**Figure 3. Adjusted $R^2$ values for a linear regression of environmental parameters against the fractional abundance (FA)**
**of brGDGT-Ia calculated within different structural sets. Colors denote the strengths of the relationships, from the**
**minimum to the maximum observed coefficients of determination, with white being the median of the dataset.**
**"Compounds" denote all brGDGTs used in the FA calculation. All values are significant (p < 0.01) unless marked with an**
**asterisk.**



## 4 Results and Discussion

### 4.1 Distributions of brGDGTs in Icelandic and Canadian lake sediments

The new Icelandic and Canadian lake sediments bolster the global dataset on the cold end, extending the lowest MAT from -0.2°C to -18°C and containing 28 of the 30 coldest samples by MAT. The low temperatures are reflected in the brGDGT distributions of these sediments (Fig. 4), which contain on average a higher $fIIIa_{Full}$ and lower $fIa_{Full}$ than the other samples in this dataset. Additionally, the Canadian and Icelandic datasets provide important end-member samples, including those with the highest $fIIIa_{Full}$, $fIIIb_{Full}$, and $fIIIc_{Full}$ and the lowest or second-lowest $fIa_{Full}$, $fIb_{Full}$, and $fIc_{Full}$. The new samples also contain some of the lowest fractional abundances of 6-methyl isomers, providing low end-member values for $fIIa'_{Full}$, $fIIb'_{Full}$, $fIIc'_{Full}$ and below-average values for the hexamethylated 6-methyl brGDGTs.

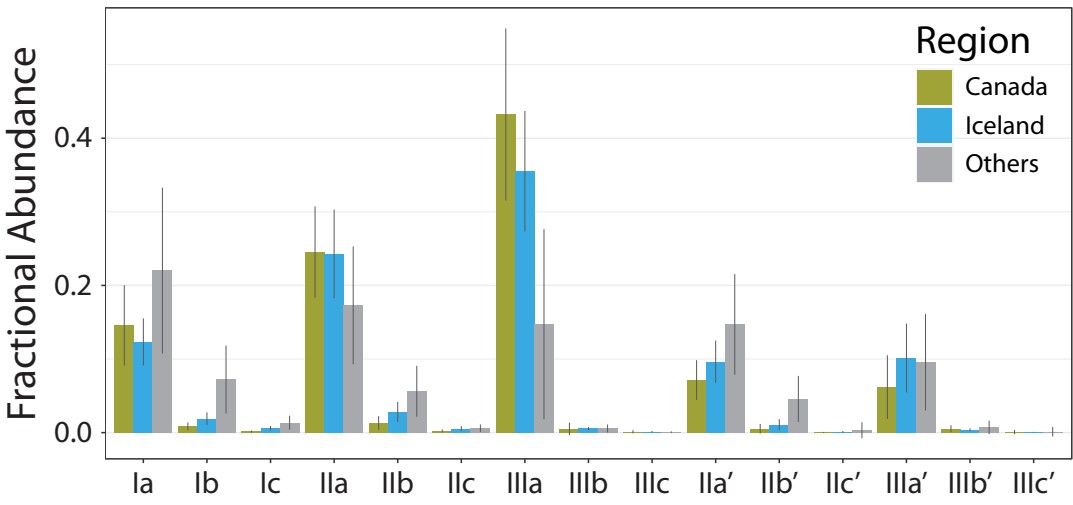

**Figure 4.** Average fractional abundances (calculated within the standard Full set) of new samples from Canada and Iceland as compared to the average of the other samples in this study (Others).

### 4.2 Temperature relationships with brGDGTs

In this section, we show that two adjustments can be made to significantly improve the correlations between brGDGT FAs and temperature in lake sediments: 1) replace MAT with a warm-season temperature index and 2) use FAs calculated within the Meth structural set. The effect of these two adjustments on one representative compound, brGDGT-Ia, is shown from left to right in Fig. 5. The relationship between temperature and fIa is improved from a weak but significant correlation with a large error ($R^2 = 0.49$; RMSE = 7.00°C) to a stronger correlation with a smaller error ($R^2 = 0.88$; RMSE = 2.42°C). The two adjustments are detailed in the following sections.





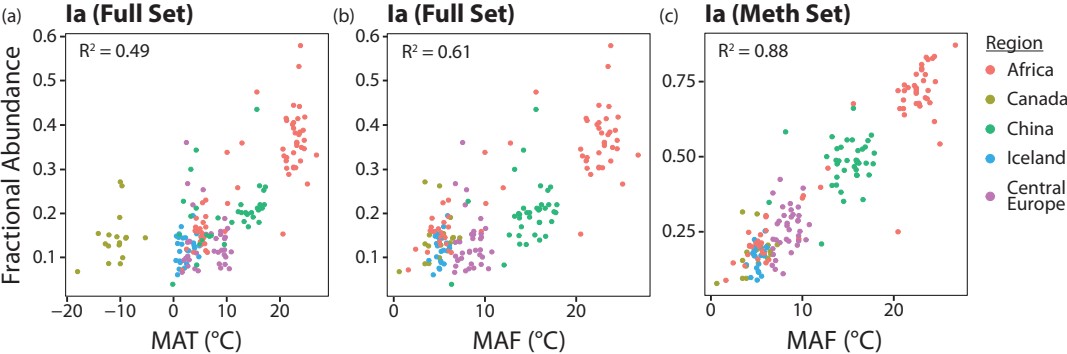

**Figure 5. From left to right, the effects of substituting the Months Above Freezing (MAF) warm-season index for Mean Annual Temperature (MAT) and fIa$_{Meth}$ for fIa$_{Full}$ on the relationship between the fractional abundance of brGDGT-Ia and temperature.**

### 4.2.1 Warm-season temperatures outperform MAT

To assess the possibility of a warm-season bias in brGDGT temperature reconstructions, we tested the relationships between brGDGTs and five air temperature indices: MAT, MAF, MST, WMT, and SWI (Sect. 2.4). The correlation between these variables and the FA of one representative compound, fIa$_{Meth}$, is shown in Fig. 6. In all cases, substituting MAT with a warm-season temperature variable draws samples with strong seasonality into the main body of data. MAF performs best for brGDGT-Ia ($R^2 = 0.88$), with high-seasonality lakes falling progressively out of alignment for SWI ($R^2 = 0.84$), MST ($R^2 = 0.64$), and WMT ($R^2 = 0.60$). This result was upheld when performing temperature calibrations; MAF outperformed all other measures of temperature examined in this study (Sect. 4.4.1).

A possible explanation for the success of the MAF temperature index is that the activity of brGDGT-producing microbes may be heavily depressed under lake ice and/or in frozen soils (Pearson et al., 2011; Shanahan et al., 2013; Peterse et al., 2014; Cao et al., 2020). However, studies employing sub-seasonal sampling of sediment traps and suspended particulate matter in two mid-latitude lakes have shown that brGDGTs are produced within the water column throughout the year, despite the presence of ice cover (Woltering et al., 2012; Loomis et al., 2014b). These and other studies employing similar sampling techniques (Hu et al., 2016; Weber et al., 2018; van Bree et al., 2020) have additionally found production of brGDGTs to be dependent on the degree and timing of lake mixing versus stratification. Though heightened biological activity may still be the underlying driver of the observed warm-season bias, these depth- and time-resolved studies paint a complex picture of brGDGT production in lakes that precludes a simple explanation. Unfortunately, as knowledge of the timing, extent, and temperature of ice cover and mixing events is lacking for the vast majority of lakes in this study, these effects cannot be tested here. We therefore stress the empirical nature of our MAF calibrations and the need for further study.



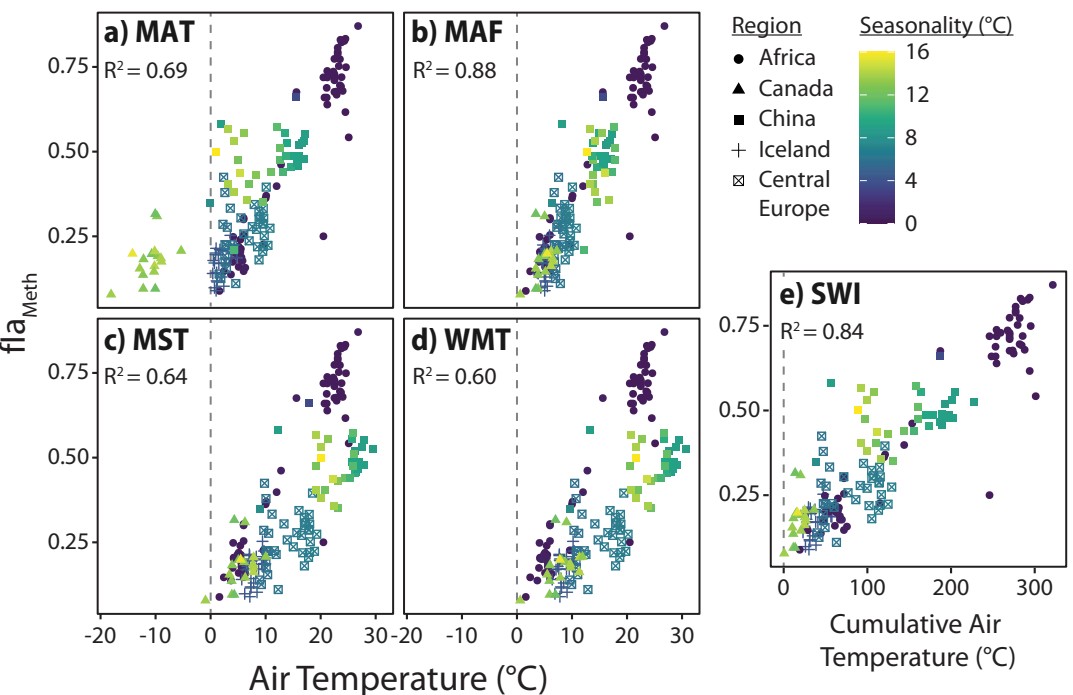


**Figure 6. Relationships between fIa$_{Meth}$ and the five air temperature variables tested in this study, a) mean annual**
**temperature (MAT), b) months above freezing (MAF), c) mean summer temperature (MST), d) warmest month**
**temperature (WMT), and e) summer warmth index (SWI). Colors indicate air temperature seasonality (standard**
**deviation of mean monthly temperatures). Dashed lines indicate mean or cumulative temperatures of 0°C for reference.**
**R² values for the relationship between fIa$_{Meth}$ and each temperature variable are provided in each subplot.**

**4.2.2 Temperature and the Methylation set**

The Methylation set provided the strongest relationships between brGDGT FAs and MAF (Fig. 7). This set

both strengthened existing correlations (e.g. fIa$_{Meth}$, Fig. 7a) and generated new ones (e.g. fIIb$_{Meth}$, Fig. 7e).
Furthermore, many relationships between FAs and temperature were revealed to be qualitatively similar regardless of
ring number. For example, the FAs of all tetramethylated compounds (fIa$_{Meth}$, fIb$_{Meth}$, fIc$_{Meth}$) had a strong positive
linear relationship with temperature (Fig. 7a-c), while those of the 5-methyl pentamethylated compounds all had a
noisier negative relationship (Fig. 7d-f). The hexamethylated FA trends were less clear in part due to their lower
abundances, but all were negatively correlated with temperature and nonlinearities were apparent (Fig. 7g-l). These
analogous trends show that the number of methylations responds similarly to temperature regardless of the number of
cyclopentane rings. Within a paleoclimate lens, this observation opens up the possibility of independent temperature
calibrations for un-, mono-, and bicyclized brGDGTs (Sect. 4.4.1). From a biological standpoint, it could imply that
methylation and cyclization play their biological roles independently, as the former appears to vary more or less freely
of the latter.





**Figure 7. Relationships between the average air temperature of months above freezing (MAF) and brGDGT FAs calculated within the Meth set. $R^2$ values are provided for each subplot, with $R^2$ values for the standard Full FAs given in parentheses for comparison. P-values were < 0.01 except where marked with an asterisk. Note that plots of IIa', IIb', and IIc' are redundant because they exactly mirror those of IIIa', IIIb', and IIIc', respectively, and are therefore not shown.**

The brGDGT temperature response appears to be agnostic to methylation position as well (Fig. 7g-i versus j-l), but only when the tetramethylated brGDGTs (Ia, Ib, and Ic) are excluded from 6-methyl FA calculations. The Meth-5Me+ subset (Fig. S3a) showed strong relationships between 5-methyl brGDGTs and MAF ($R^2 \le 0.88$; Fig. S4). On the other hand, the analogous Meth-6Me+ subset (Fig. S3c) was broadly uncorrelated with MAF ($R^2 \le 0.29$; Fig. S5). At present, there is no known mechanism whereby an additional methylation at the C5 position would have an influence on membrane physiology in a way that a methylation at C6 would not. At first glance, then, the markedly different responses of the Meth-5Me+ and Meth-6Me+ subsets to changes in temperature do not appear to support a physiological basis for the empirical relationship between temperature and brGDGT methylation number. However, when tetramethylated brGDGTs were excluded from the FA calculations for these compounds (Meth-5Me and Meth-6Me subsets; Figs. S3b and d), statistically significant and qualitatively similar temperature relationships did become visible for both isomer types (Figs. S6-7). An analogous result was found for the MC set (Figs. S3g-j and S8-11). It is not clear at this time why the inclusion of tetramethylated brGDGTs improved temperature correlations for 5-methyl compounds but weakened them for 6-methyl compounds. The discrepancy may imply one or a combination of the following: 1) the isomers are produced by different organisms; 2) the isomers serve distinct biological functions, either in addition to or apart from a temperature response; or 3) the currently-measured tetramethylated brGDGTs (Ia, Ib, and Ic) are not the precursors of 6-methyl brGDGTs. Regardless, this result allows us to combine the higher-performing Meth-5Me+ and Meth-6Me subsets to generate the Meth set (Fig. 2a), which maximizes the temperature responses of all 15 commonly-measured brGDGT (Fig. 7).

While the Meth set highlights the relationships between brGDGT FAs and temperature, it simultaneously weakens those with other environmental variables. FAs calculated in the standard Full set contain conductivity and pH dependencies ($R^2 \le 0.66$ and 0.50, respectively; Figs S25 and S32) that are greatly reduced in the Meth set ($R^2 \le 0.40$ and 0.28, respectively; Figs S19 and S26). This is evidence that many of the conductivity and pH relationships visible in the Full FAs are in fact due to the mathematical mixing of brGDGTs with different cyclization numbers and isomer designations. Holding these variations constant in the Meth FAs largely removes the effects of these environmental variables (e.g. fIIa$_{Meth}$ in Fig. S19d ). DO dependencies are weak in both the Full and Meth sets, but slightly weaker in the latter ($R^2 \le 0.40$ and 0.35, respectively, Figs S39 and S33). The Methylation set thus improves compound-specific correlations with temperature while decreasing their dependencies on other environmental variables.

**4.3 Conductivity and pH relationships with brGDGTs**

While pH is the traditional secondary target of brGDGT calibrations after temperature, numerous works have suggested that conductivity plays an important role in controlling brGDGT distributions (Tierney et al., 2010; Shanahan et al., 2013). The two variables often plot nearly colinearly in principal component analyses (Shanahan et al., 2013; Dang et al., 2018; Russell et al., 2018), suggesting that they may have similar influences on brGDGT





distributions. In our dataset, conductivity and pH were moderately correlated ($R^2 = 0.57$) and elicited similar responses
in brGDGT FAs. We therefore discuss them together in this section, with an emphasis on the more strongly-correlated
of the two, conductivity.

**4.3.1 The Isomer set and conductivity**
Conductivity provided the strongest compound-specific correlations with brGDGT FAs after temperature.
Of the basic brGDGT sets (Meth, Cyc, and Isom), the Isom set had the highest statistical performance (Fig. 8). The
FAs of all 6-methyl brGDGTs showed a positive linear correlation with conductivity in this set, with coefficients of
determination as high as $R^2 = 0.70$ (Fig. 8d). Furthermore, this positive correlation was broadly independent of both
methylation number and cyclization number, indicating that methylation position varies with conductivity irrespective
of other structural properties. The 6-methyl brGDGTs were also all positively correlated with pH, but more weakly so
($R^2 \leq 0.54$, Fig. S28). A relationship between brGDGT isomers and pH in lake sediments has been previously observed
and quantified by the isomerization of branched tetraethers (IBT, Eq. A14; Ding et al., 2015) and the Isomer Ratio of
6-methyl isomers ($IR_{6Me}$, Eq. A8; Dang et al., 2016). However, these indices were more closely tied to conductivity
(IBT $R^2 = 0.65$, $IR_{6Me}$ $R^2 = 0.66$) than pH (IBT $R^2 = 0.55$, $IR_{6Me}$ $R^2 = 0.49$) in our dataset. These results indicate that
isomer abundances are primarily dependent on conductivity, but have some relation to pH as well. Temperature
correlations were also present in the Isom subset ($R^2 \leq 0.57$ with MAF, Fig. S14) that were stronger than the inherent
correlation between MAF and conductivity in our dataset ($R^2 = 0.48$). While conductivity is the primary control on
isomer ratios, temperature may therefore play a secondary role. DO provided little to no correlation in the Isom set
($R^2 \leq 0.38$, Fig. S35).



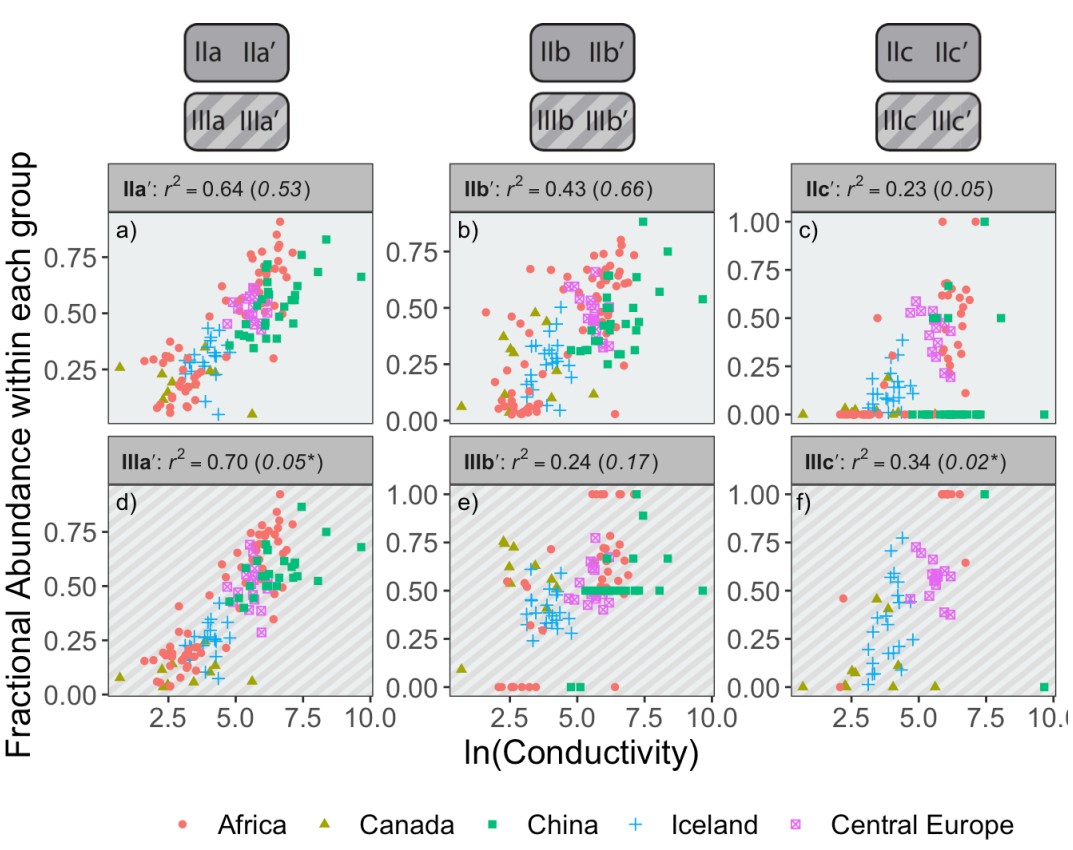


**Figure 8. Relationships between the natural logarithm of conductivity and brGDGT FAs calculated within the Isom set. R² values are provided for each subplot, with R² values for the standard Full FAs given in parentheses for comparison. P-values were < 0.01 except where marked with an asterisk. Note that plots of 5-methyl compounds are redundant because they exactly mirror those of their 6-methyl counterparts; only the 6-methyl FAs are shown.**

### 4.3.2 The Cyclization set and pH

The Cyclization set highlighted the relationship between brGDGTs and pH. As pH increased, all cyclized brGDGTs were found in greater relative abundance (Fig. 9). This result reinforces previous observations that higher ring number is associated with higher pH, as has been quantified by the CBT (Weijers et al., 2007), #rings$_{tetra}$ (Sinninghe Damsté, 2016), DC (Baxter et al., 2019), and related indices (see Appendix). However, this pH relationship has thus far been demonstrated primarily in soils; the single-compound FAs of the Cyc subset already provide the strongest pH correlations ($R^2$ = 0.61, Fig. 9b) yet reported in lake sediments. The Cyc set also reveals that compounds with structural similarities exhibit analogous responses to pH. For example, all monocyclized brGDGTs show a nonlinear increase with pH (Fig. 9b, e, h, k, n), while uncyclized brGDGTs all exhibit a nonlinear decrease (Fig. 9a, d, g, j, m). These trends are apparent regardless of methylation number or methylation position, suggesting that ring number is broadly independent of both. This independence may imply that alkyl-chain cyclization serves its biological



function(s) regardless of the number and position of methylations present. It also allows for the construction of
independent pH calibrations for tetra-, penta-, and hexamethylated brGDGTs (Sect. 4.4.2).

Though the Cyc set FAs were most strongly correlated with pH ($R^2 \leq 0.61$), they also exhibited robust

relationships with conductivity ($R^2 \leq 0.57$; Fig. S20). All cyclized compounds showed positive correlations with
conductivity, and this increase was largely independent of methylation number or position. These results indicate that
ring number is primarily dependent on pH, but is correlated with conductivity in a similar manner. The brGDGT
indices showed an analogous result; all cyclization indices (CBT and related indices, #rings$_{tetra}$ and related indices,
and DC; see Appendix) correlated most strongly with pH, but also exhibited weaker relationships with conductivity.
The Cyc set exhibited little to no correlation with either temperature ($R^2 \leq 0.25$) or DO ($R^2 \leq 0.08$), suggesting that
neither of these environmental variables plays an important role in controlling brGDGT cyclization.



Figure 9. Relationships between pH and brGDGT FAs calculated within the Cyc set. R² values are provided for each subplot, with R² values for the standard Full FAs given in parentheses for comparison. P-values were < 0.01 except where marked with an asterisk.



### 4.3.3 The Combined Cyclization-Isomer set strengthens both conductivity and pH trends

Though the strongest conductivity trends were displayed by the Isom set ($R^2 \leq 0.70$), correlations were also present in the Cyc FAs ($R^2 \leq 0.57$, Fig. S20). Similarly, the Cyc set contained the highest pH dependencies ($R^2 \leq 0.61$), but notable relationships were visible in the Isom set as well ($R^2 \leq 0.54$, Fig. S28). To take advantage of all of these conductivity and pH relationships, we therefore used the combined Cyc-Isom set, which holds only methylation number constant while allowing both cyclization number and methylation position to vary (Fig. 2). Both conductivity and pH trends were strengthened in this combination set ($R^2 \leq 0.73$ and $0.62$, Fig. S23 and S30, respectively), especially for the uncyclized compounds. However, temperature correlations were also increased in the CI set ($R^2 \leq 0.60$, Fig. S16), a potentially convoluting influence that may not be desired.

### 4.4 Calibrations

For each set, combined set, and subset defined in Sect. 3, we performed linear and quadratic regressions against temperature, conductivity, pH, dissolved oxygen, and lake geometry variables using SFS/SBE and combinatoric fitting methods. We found temperature and conductivity to provide the strongest empirical calibrations with brGDGTs, followed by pH and dissolved oxygen, and discuss our recommended calibrations below.

### 4.4.1 Temperature Calibrations

We performed regressions against five temperature variables to generate multiple global-scale calibrations. Of the temperature indices that we tested, MAF provided the fit with the highest statistical significance ($R^2 = 0.91$), followed closely by MST ($R^2 = 0.90$), SWI ($R^2 = 0.89$), and WMT ($R^2 = 0.88$). MAT provided calibrations with high statistical performance as well ($R^2 \leq 0.87$). However, these fits showed clear seasonality biases in their residuals, resulting in substantial over-estimations of MAT for cold sites (Fig. S40). Furthermore, they often relied heavily on low-abundance compounds as fitting variables (IIc, IIb', IIIc, IIIc', IIIb, and IIIb'). We therefore do not recommend a brGDGT MAT calibration and focus our discussion on the warm-season temperature indices, especially MAF.

Methylation number was the single most important structural variable for temperature calibrations. The Meth set, which allowed only methylation number to vary, provided a MAF calibration ($R^2 = 0.90$, Fig. 10b) that was on par with other recent global and regional lake sediment calibrations ($R^2 = 0.85$ to $0.94$; Dang et al., 2018; Russell et al., 2018; Martínez-Sosa et al., 2020b). The MI, MC, and Full sets, which additionally allowed for changes in cyclization number and/or methylation position, added little to the calibration performance ($R^2 = 0.90$ to $0.91$, Fig. 10a). Furthermore, sets which held methylation number constant – Cyc, Isom, and CI – performed markedly worse ($R^2 = 0.51$, $0.63$, and $0.67$, respectively, Fig. 10a). These results indicate that effectively all of the temperature dependence of the 15 commonly-measured brGDGTs is captured by methylation number alone.



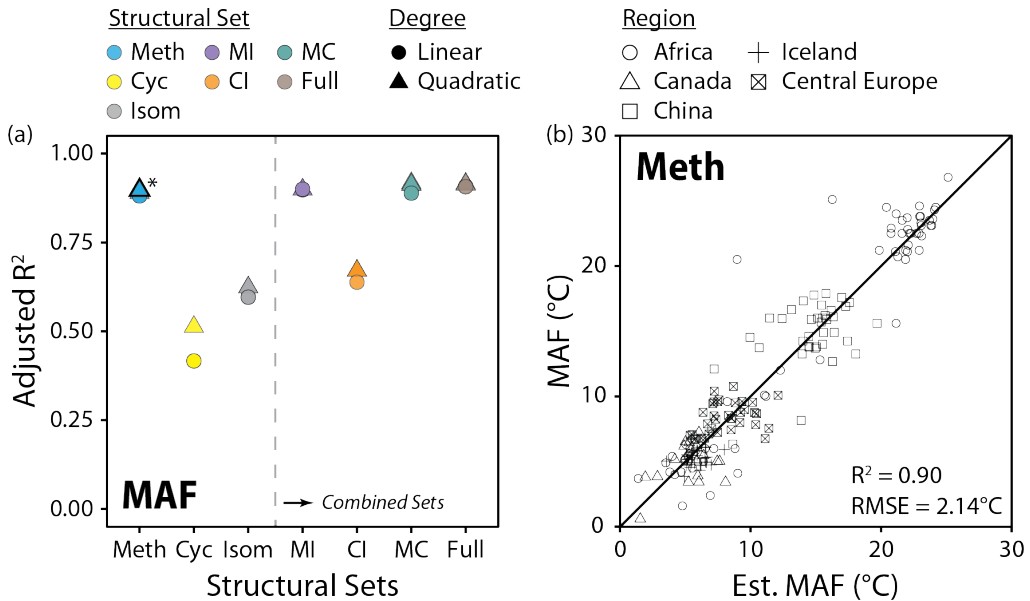

**Figure 10. a)** Performance (adjusted $R^2$) of all linear and quadratic fits for the mean air temperature of months above freezing (MAF) and brGDGT fractional abundances (FAs) calculated within the basic (Meth, Cyc, Isom; left of dashed line) and combined (Meth-Isom (MI), Cyc-Isom (CI), Meth-Cyc (MC), and Full sets; right of dashed line) structural sets. Results of both the SFS/SBE and combinatoric fitting methods are plotted. The fit we suggest for general use (Meth set, quadratic, SFS/SBE; Eq. 10) is bolded and marked with an asterisk in a) and plotted in b). "Est. MAF" is the MAF temperature estimated using this suggested fit.

Within the Meth set, independent calibrations were generated for the Meth$_a$, Meth$_b$, and Meth$_c$ subsets as well. These subsets consist of only un-, mono-, and bicylcized compounds, respectively (Fig. 2a). As the FAs of these subsets were calculated independently, we were able to test for temperature calibrations of each subset alone. The Meth$_a$ subset provided the strongest of the subset MAF fits ($R^2 = 0.88$). This calibration has the notable advantage of employing only the 3 most abundant brGDGTs typically found in nature, Ia, IIa, and IIIa, which may allow for temperatures to be reproduced with high fidelity from even organic-lean samples. Furthermore, since it uses only non-cyclized, 5-methyl brGDGTs, it may be less subject to influence by the environmental factors that impact cyclization numbers and isomer ratios. MAF calibrations were also obtained using only the monocyclized (Meth$_b$, $R^2 = 0.79$) and bicyclized (Meth$_c$, $R^2 = 0.74$) brGDGTs. These fits represent, to our knowledge, the first calibrations that make use of only cyclized brGDGTs. This is a noteworthy result as it shows conclusively what can be seen by eye in Fig. 7; the relationship between temperature and methylations is a broad feature of brGDGTs that is present regardless of the number of rings on the carbon backbone. These un-, mono-, and bicyclized calibrations may find use in the case that one or more brGDGTs are suspected to be influenced by variables other than temperature.

Of the brGDGT temperature indices that we tested (see Appendix), MBT'$_{5Me}$ performed best. This index correlated better with MAF ($R^2 = 0.89$) than any other warm-season variable (SWI $R^2 = 0.84$; MST $R^2 = 0.70$; MAT $R^2 = 0.70$; WMT $R^2 = 0.66$). The slope and intercept of the MAF/MBT'$_{5Me}$ calibration (MAF = -0.5 ($\pm$ 0.4) + 30.4 ($\pm$ 0.8) * MBT'$_{5Me}$) were similar to the MAT/MBT'$_{5Me}$ calibration presented by Russell et al. (2018) (MAT = -1.21 + 32.42 * MBT'$_{5Me}$). This may suggest that MBT'$_{5Me}$-derived temperatures using the Russell et al. (2018) calibration in





cold regions are best considered to reconstruct MAF rather than MAT. Though MBT'$_{6Me}$ was previously found to
correlate well with temperature on a regional scale ($R^2$ = 0.75; Dang et al., 2018), it was not correlated with any
temperature variable in our global dataset ($R^2 \leq 0.12$). Finally, we note that the Community Index (De Jonge et al.,
2019), which was associated with bacterial community changes in geothermally heated Icelandic soils, is identical to
our fIa$_{Meth}$. This may suggest that the strong connection between fIa$_{Meth}$ and MAF could also be driven at least in part
by changes in microbial community composition. However, genomic data is not currently available for the majority
of the sites in this study to test this hypothesis.

### 548     4.4.2 Conductivity and pH Calibrations

Conductivity outperformed pH in our calibrations ($R^2$ = 0.83 versus 0.74) and was the second most important
predictor of brGDGT distributions in our dataset after temperature. Both cyclization number and methylation position
were important to the success of these calibrations. The Cyc set, which allowed only cyclization number to change,
provided a conductivity fit with $R^2$ = 0.73 (Fig. 11a). The Isom set, which isolated trends in isomer abundances,
generated a slightly stronger calibration with $R^2$ = 0.76. When both of these structural properties were allowed to vary
together in the CI set, the calibration was markedly improved ($R^2$ = 0.83, Fig. 11b). In contrast, methylation number
was a poorer predictor of conductivity alone (Meth set $R^2$ = 0.65) and did not improve upon the CI correlation in the
combined sets (MI $R^2$ = 0.80; MC $R^2$ = 0.83; Full $R^2$ = 0.81). The CI$_{III}$ and CI$_{II}$ subsets also provided conductivity
calibrations with relatively high statistical performance ($R^2$ = 0.75, 0.73). The CI$_I$ subset performed worse ($R^2$ = 0.65),
likely due to the fact that no isomer variations are present in these FAs. These subset-specific fits are the first of their
kind, and their success indicates that the relationship between brGDGTs and conductivity is present regardless of
methylation number.
In addition to conductivity, the CI set also best captured the relationship between brGDGTs and pH ($R^2$ =
0.73; Fig. 11c and d). The addition of methylation number variations in the Full set did not substantially improve the
calibration ($R^2$ = 0.74), indicating that the majority of the relationship between brGDGTs and pH is captured by
cyclization number and methylation position. Of these two structural variables, cyclization number was more
important for pH; the Cyc set calibration ($R^2$ = 0.67) outperformed those from the Isom and Meth sets ($R^2$ = 0.59 for
both). The CI$_I$, CI$_{II}$, and CI$_{III}$ subsets provided weaker, but significant calibrations with comparable performances to
one another ($R^2$ = 0.67, 0.68, and 0.62, respectively), indicating again that pH relationships are more or less
independent of methylation number.



Figure 11. Performance (adjusted R²) of all linear and quadratic fits for lake water conductivity (a) and pH (c) and brGDGT fractional abundances (FAs) calculated within the basic (Meth, Cyc, Isom; left of dashed lines) and combined (Meth-Isom (MI), Cyc-Isom (CI), Meth-Cyc (MC), and Full sets; right of dashed lines) structural sets. Results of both the SFS/SBE and combinatoric fitting methods are plotted. The fits we suggest for general use (CI set, quadratic, combinatoric, for both variables; Eqs. 12-13) is bolded and marked with an asterisk in a) and c) and plotted in b) and d). "Est. ln(Conductivity)" and "Est. pH" are the natural logarithm of lake water conductivity and pH estimated using these suggested fits.





### 4.4.3 Dissolved Oxygen and Lake Geometry Calibrations

There is increasing evidence that oxygen availability strongly affects lacustrine brGDGT distributions (Colcord et al., 2017; Weber et al., 2018; van Bree et al., 2020; Yao et al., 2020). We therefore tested for calibrations within our dataset with mean and minimum dissolved oxygen concentration ($DO_{mean}$ and $DO_{min}$). As lake morphology can be an important predictor of lake oxygen levels (Hutchinson, 1938; Nürnberg, 1995), we also tested the natural logarithms of maximum water depth (Depth), the ratio of lake surface area to maximum depth (SA/D), and approximate lake volume.

None of the DO or lake geometry variables generated strong brGDGT calibrations ($R^2 \leq 0.63$; Table S1). The highest-performing fit was provided by the Meth set with $DO_{mean}$ ($R^2 = 0.63$, Fig. S41). Moderate correlations were found with $DO_{min}$ as well (Cyc set, $R^2 = 0.55$). Lake depth alone was a poor predictor of brGDGT distributions ($R^2 = 0.35$), but both volume and the ratio of surface area to depth were found to provide moderate correlations ($R^2 = 0.55$ and $0.59$, respectively). However, none of these lake morphology variables was itself well-correlated with $DO_{mean}$ or $DO_{min}$ ($R^2 \leq 0.22$) in this dataset, and we therefore cannot explain their relationship with brGDGTs at this time. Additionally, although the HP5 index (Eq. A13) was recently shown to reflect redox conditions via a correlation with lake water depth (Yao et al., 2020), it does not correlate with any of our lake geometry indices ($R^2 \leq 0.02$) and only weakly correlates with DO ($R^2 \leq 0.28$) in this dataset, indicating that it may be primarily useful for within-lake studies.

The Meth set provided both the strongest $DO_{mean}$ and MAF calibrations, raising the possibility that DO may have a problematic influence on that calibration's temperature estimates. Individual Meth FAs were weakly correlated with $DO_{mean}$ at best ($R^2 \leq 0.35$), however, and the residuals of the Meth/MAF fit in Eq. 10 showed no correlation with $DO_{mean}$ ($R^2 = 0.01$, p = 0.2). Given these weak relationships, we do not see evidence for the influence of DO on temperatures reconstructed with the Meth calibration in our dataset.

In light of increasing evidence that oxygen availability strongly affects lacustrine brGDGT distributions, it is perhaps surprising that we do not find a stronger correlation between brGDGTs and DO. However, the effects of DO on brGDGT distributions appear to be highly site-specific. For example, some detailed studies have found elevated levels of brGDGT-IIIa in low oxygen conditions (Weber et al., 2018; Yao et al., 2020), but another found all of the most common brGDGTs *except* IIIa in abundance in the oxygen-depleted hypolimnion (van Bree et al., 2020). A third detailed study found no correlation between brGDGTs and oxygen at all (Loomis et al., 2014b), and no calibration study to date has found regional or global trends. The wide range of possible drivers of DO – mixing regimes, eutrophication state, and ice cover, to name a few – may play a role in the incoherent relationship between brGDGTs and DO in these studies and our own. Additionally, DO measurements are often taken at the time of sampling and are most likely not representative of the annual range. For the Canadian and Icelandic lakes in this study, for example, DO was depleted under lake ice relative to ice-free conditions in 10 out of 11 cases. Few of the lakes in our study have continuous DO monitoring data available, and most are from Central Europe. Therefore, while our study does find significant correlations between brGDGTs and DO, we do not recommend a calibration for general use and instead highlight the need for further study.



### 4.4.4 Recommended Calibrations

The Meth, MC, MI, and Full subsets all provided MAF temperature calibrations with comparable $R^2$ (0.90 to 0.91) and RMSE (1.97 to 2.14°C) values (Fig. 10a, Table S1). However, the Meth set provided the FAs with the strongest compound-specific relationships with MAF ($R^2 \leq 0.88$) in the modern dataset and exhibited little influence ($R^2 \leq 0.49$) from any other variable examined in this study. In contrast, the MC, MI, and Full sets all exhibited stronger relationships with pH and conductivity (Fig. 11a and c). We therefore recommend the highest-performing Meth calibration (Fig. 10b; Eq. 10; n = 182, $R^2$ = 0.90, RMSE = 2.14°C) for general use in lake sediments:

$$MAF\ (°C) = 92.9(\pm 15.98) + 63.84(\pm 15.58) \times fIb_{Meth}^2 - 130.51(\pm 30.73) \times fIb_{Meth}$$
$$- 28.77(\pm 5.44) \times fIIa_{Meth}^2 - 72.28(\pm 17.38) \times fIIb_{Meth}^2 - 5.88(\pm 1.36) \times fIIc_{Meth}^2$$
$$+ 20.89(\pm 7.69) \times fIIIa_{Meth}^2 - 40.54(\pm 5.89) \times fIIIa_{Meth}$$
$$- 80.47(\pm 19.19) \times fIIIb_{Meth} \tag{10}$$

The Full set MAF calibration provided the highest $R^2$ and lowest RMSE in our dataset. This fit may be applicable in settings with good conductivity or pH control and may be useful for comparison with previous calibrations. It is therefore provided in Eq. (11) (n = 182, $R^2$ = 0.91, RMSE = 1.97°C):

$$MAF\ (°C) = -8.06(\pm 1.56) + 37.52(\pm 2.35) \times fIa_{Full} - 266.83(\pm 98.61) \times fIb_{Full}^2$$
$$+ 133.42(\pm 19.51) \times fIb_{Full} + 100.85(\pm 9.27) \times fIIa'^2_{Full} + 58.15(\pm 10.09) \times fIIIa'^2_{Full}$$
$$+ 12.79(\pm 2.89) \times fIIIa_{Full} \tag{11}$$

The calibrations presented in Eqs. (10) and (11) allow for the quantitative reconstruction of warm-season air temperatures from lake sediment archives, including those at high latitudes. The statistical performance of these fits is comparable to recently-published calibrations (Russell et al. (2018): $R^2$ = 0.94, RMSE = 2.14°C, n = 65; Martínez-Sosa et al. (2020): $R^2$ = 0.85, RMSE = 2.8°C, n = 261; Dang et al. (2018): $R^2$ = 0.91, RMSE = 1.10°C, n = 39). Subset-specific calibrations (Meth$_a$, Meth$_b$, and Meth$_c$) are also statistically comparable and are available in the Supplement (Eq. S1-3). We do not recommend an independent MAT calibration due to residual seasonality biases (Fig. S40), though we note that MAT is often identical to MAF in warm or low-seasonality settings.

The CI set generated the highest-performing conductivity calibration ($R^2$ = 0.83, RMSE = 0.66; Fig. 11b, Table S1). The Full, MC, and MI sets provided calibrations that were statistically comparable ($R^2$ = 0.80 to 0.83, RMSE = 0.65 to 0.70), but their FAs contained marked temperature dependencies ($R^2 \leq 0.70$, 0.77, and 0.75, respectively; Figs S18 , S17 and S15). We therefore recommend the top CI fit for use (Eq. 12; n = 143, $R^2$ = 0.83, RMSE = 0.66), and provide subset-specific calibrations for CI$_I$, CI$_{II}$, and CI$_{III}$ in the Supplement (Eq. S4-6):

$$ln(Cond.) = 6.62(\pm 1.01) + 8.87(\pm 1.24) \times fIb_{CI} + 5.12(\pm 1.54) \times fIIa'^2_{CI} + 10.64(\pm 1.88) \times fIIa_{CI}^2$$
$$- 8.59(\pm 2.21) \times fIIa_{CI} - 4.32(\pm 1.46) \times fIIIa'^2_{CI} - 5.31(\pm 0.95) \times fIIIa_{CI}^2$$
$$- 142.67(\pm 36.08) \times fIIIb_{CI}^2 \tag{12}$$

A brGDGT-based paleoconductivity reconstruction has yet to be attempted, but diatom-inferred conductivity records show the potential for this variable to provide valuable insight into changes in lake hydrology. These records have reconstructed changes in precipitation and evaporation balance, lake level fluctuations, meltwater influx events, and the isolation of a lake from the sea (Ng and King, 1999; Yang et al., 2004; Stager et al., 2013). The conductivity



calibration in Eq. (12) thus enables brGDGTs to be tested as a new alternative or complementary proxy in
paleohydrology reconstructions.

The Full and CI sets provided calibrations with pH that were statistically comparable to one another ($R^2$ =
0.74 and 0.73, Fig. 11c, Table S1). Given the stronger temperature dependencies of the Full set, we recommend the
CI calibration (Fig. 11d; Eq. 13; n = 154, $R^2$ = 0.73, RMSE = 0.57) for use and provide its subset-specific calibrations
($CI_I$, $CI_{II}$, and $CI_{III}$) in the Supplement (Eq. S7-9):

$$pH = 8.93(\pm 0.21) - 3.84(\pm 0.25) \times fIa_{CI}^2 + 2.63(\pm 0.35) \times fIIa'_{CI} \qquad (13)$$

The residuals of the fit in Eq. (13) have a weak but significant correlation with pH ($R^2$ = 0.17), causing it to
overestimate pH for acidic samples and underestimate it for alkaline ones. A similar bias in pH calibrations has been
previously observed in soils (De Jonge et al., 2014a). We therefore caution the use of this calibration in acidic (pH <
5) or alkaline (pH > 9) conditions.

Previous work has demonstrated the value of brGDGT-derived pH records in studies of terrestrial
paleoclimate (Tyler et al., 2010; Cao et al., 2017; Fastovich et al., 2020). However, these studies relied on calibrations
generated from soils and/or analyses in which the 5- and 6-methyl isomers were not separated. The fit presented in
Eq. (13) may improve such studies by providing a globally-distributed pH calibration in lake sediments using the latest
chromatographic methods which improves upon the error of previously available calibrations (RMSE = 0.80, Russell
et al., 2018).

Dissolved oxygen and lake geometry calibrations generated significant, but statistically weaker fits ($R^2 \leq$
0.63). Due to the low $R^2$ of these calibrations and an incomplete understanding of the relationship between DO and
brGDGT distributions, we do not recommend their application at this time. However, the equation for the highest-
performing variable, $DO_{mean}$, is provided in the Supplement for reference (Eq. S10).
**5 Conclusions**

We have shown that brGDGT structural sets and warm-season temperature indices improve correlations with
environmental parameters while advancing our biological understanding of the lipids themselves. Grouping brGDGTs
into structural sets based on methylation number, methylation position, and cyclization number elucidated the
relationships between environmental variables and brGDGT structures. These sets revealed that methylation number
fully captures the relationship between brGDGT distributions and temperature. They also showed the relative
abundance of 5- and 6-methyl isomers to be dependent on conductivity and cyclization number to be primarily tied to
pH. The deconvolved relationships provided by these subsets allowed for the generation of calibrations with
temperature and pH that relied on fewer compounds with robust modern trends in a global dataset. They additionally
revealed conductivity to be the second-most important variable in controlling brGDGT distributions and provided a
calibration for this oft-overlooked variable, which may find use as a proxy for precipitation/evaporation balances or
hydrologic changes.

The structural sets also provided insight into the biological underpinnings of brGDGT structural diversity.
The Meth, Cyc, and Isom sets gave evidence that methylation number, cyclization number, and methylation position
vary more or less independently of one another across environmental gradients. They further revealed that the
inclusion of tetramethylated compounds (Ia, Ib, Ic) enhances the temperature dependencies of 5-methyl compounds,



but erases those of their 6-methyl counterparts. As the microbial producers of brGDGTs have yet to be identified and
cultured, the structural set approach thus provides a valuable tool for investigating controls on brGDGT diversity with
a biological lens.

Warm-season temperatures outperformed MAT as the most important predictors of brGDGT distributions.

We introduced 43 new lake sediment samples from sites with low MAT and high seasonality. In conjunction with a
global dataset, these samples showed a clear warm-season bias in brGDGT temperature relationships, with MAF
providing the strongest fits in our dataset. The warm-season bias may suggest a direct or indirect connection to
heightened primary productivity in the summer. Alternatively, it may be the result of a more complex relationship
with dynamic lake processes such as mixing events. While further study is needed to unravel these complications, the
strong empirical calibrations presented here support the use of the brGDGT paleotemperature proxy to quantitatively
reconstruct warm-season air temperatures from high-latitude lake sediments.

In summary, the use of brGDGT structural sets and warm-season temperature indices deconvolved

relationships between brGDGT structure and environmental gradients, revealed trends with biological implications,
and tied brGDGT distributions to warm-season temperatures. Furthermore, they allowed for the construction of
improved temperature and pH calibrations as well as the first brGDGT conductivity calibration in a global lake
sediment dataset. Our results thus allow brGDGTs to be used to quantitatively reconstruct warm-season air
temperatures and lake water conductivity and pH from lake sediment archives and provide a new methodology for the
study of brGDGTs in the future.
**Appendix**


**Figure A1. Structures of the 15 commonly-measured brGDGTs along with their schematic representations, with C6**
**methylations denoted in red in the schematics.**

Full and schematic structures of the 15 commonly-measured brGDGTs are provided in Fig. A1. Table A1

details equations for calculating FAs within the structural sets.





| (Sub)set name | (Sub)set compounds | Fractional abundance equation |
|---|---|---|
| Full | S = {Ia, Ib, Ic, IIa, IIb, IIc, IIIa, IIIb, IIIc, IIa', IIb', IIc', IIIa', IIIb', IIIc'} | $fxy_{[(sub)set\ name]} = xy / \sum S$ |
| MC-5Me+ | S = {Ia, Ib, Ic, IIa, IIb, IIc, IIIa, IIIb, IIIc} | |
| MC-6Me+ | S = {Ia, Ib, Ic, IIa', IIb', IIc', IIIa', IIIb', IIIc'} | |
| MC-5Me | S = {IIa, IIb, IIc, IIIa, IIIb, IIIc} | |
| MC-6Me | S = {IIa', IIb', IIc', IIIa', IIIb', IIIc'} | |
| MC | *Use MC-5Me+ and MC-6Me FAs* | |
| MI | $S_a$ = {Ia, IIa, IIIa, IIa', IIIa'} $S_b$ = {Ib, IIb, IIIb, IIb', IIIb'} $S_c$ = {Ic, IIc, IIIc, IIc', IIIc'} | $fxy_{[(sub)set\ name]} = xy / \sum S_y$ |
| Meth-5Me+ | $S_a$ = {Ia, IIa, IIIa} $S_b$ = {Ib, IIb, IIIb} $S_c$ = {Ic, IIc, IIIc} | |
| Meth-6Me+ | $S_a$ = {Ia, IIa', IIIa'} $S_b$ = {Ib, IIb', IIIb'} $S_c$ = {Ic, IIc', IIIc'} | |
| Meth-5Me | $S_a$ = {IIa, IIIa} $S_b$ = {IIb, IIIb} $S_c$ = {IIc, IIIc} | |
| Meth-6Me | $S_a$ = {IIa', IIIa'} $S_b$ = {IIb', IIIb'} $S_c$ = {IIc', IIIc'} | |
| Meth | *Use Meth-5Me+ and Meth-6Me FAs* | |
| CI | $S_I$ = {Ia, Ib, Ic} $S_{II}$ = {IIa, IIb, IIc, IIa', IIb', IIc'} $S_{III}$ = {IIIa, IIIb, IIIc, IIIa', IIIb', IIIc'} | $fxy_{[(sub)set\ name]} = xy / \sum S_x$ |
| Cyc-5Me | $S_I$ = {Ia, Ib, Ic} $S_{II}$ = {IIa, IIb, IIc} $S_{III}$ = {IIIa, IIIb, IIIc} | |
| Cyc-6Me | $S_I$ = {Ia, Ib, Ic} $S_{II}$ = {IIa', IIb', IIc'} $S_{III}$ = {IIIa', IIIb', IIIc'} | |
| Cyc | *Use Cyc-5Me and Cyc-6Me FAs* | |
| Isom | $S_{IIa}$ = {IIa, IIa'}; $S_{IIIa}$ = {IIIa, IIIa'} $S_{IIb}$ = {IIb, IIb'}; $S_{IIIb}$ = {IIIb, IIIb'} $S_{IIc}$ = {IIc, IIc'}; $S_{IIIc}$ = {IIIc, IIIc'} | $fxy_{[(sub)set\ name]} = xy / \sum S_{xy}$ |


**Table A1. Equations for calculating FAs within brGDGT subsets, where f*xy* and *xy* are the fractional and absolute**
**abundances of the 5- or 6-methyl brGDGT with Roman numeral *x* (I, II, or III) and alphabet letter *y* (a, b, or c).**
We used the following previously-defined brGDGT indices in this study: CBT (Weijers et al., 2007); MBT'
(Peterse et al., 2012); MBT'$_{5Me}$, MBT'$_{6Me}$, CBT$_{5Me}$, CBT', and Index1 (De Jonge et al., 2014a); IR$_{6Me}$ (Dang et al.,
2016); #rings$_{tetra}$, #rings$_{penta\ 5Me}$, and #rings$_{penta\ 6Me}$ (Sinninghe Damsté, 2016); Degree of Cyclization (DC) as
reformulated by (Baxter et al., 2019); HP5 (Yao et al., 2020), isomerization of branched tetraethers (IBT; Ding et al.,
2015), community index (CI; De Jonge et al., 2019). Their equations are given below:





$$CBT = -log\left(\frac{Ib + IIb + IIb'}{Ia + IIa + IIa'}\right) \tag{A1}$$

$$MBT' = \frac{(Ia + Ib + Ic)}{(Ia + Ib + Ic + IIa + IIb + IIc + IIIa + IIa' + IIb' + IIc' + IIIa')} \tag{A2}$$

$$MBT'_{5Me} = \frac{(Ia + Ib + Ic)}{(Ia + Ib + Ic + IIa + IIb + IIc + IIIa)} \tag{A3}$$

$$MBT'_{6Me} = \frac{(Ia + Ib + Ic)}{(Ia + Ib + Ic + IIa' + IIb' + IIc' + IIIa')} \tag{A4}$$

$$CBT_{5Me} = -log\left(\frac{Ib + IIb}{Ia + IIa}\right) \tag{A5}$$

$$CBT' = -log\left(\frac{Ic + IIa' + IIb' + IIc' + IIIa' + IIIb' + IIIc'}{Ia + IIa + IIIa}\right) \tag{A6}$$

$$Index1 = log\left(\frac{Ia + Ib + Ic + IIa' + IIIa'}{Ic + IIa + IIc + IIIa + IIIa'}\right) \tag{A7}$$

$$IR_{6Me} = \frac{(IIa' + IIb' + IIc' + IIIa' + IIIb' + IIIc')}{(IIa' + IIb' + IIc' + IIIa' + IIIb' + IIIc' + IIa + IIb + IIc + IIIa + IIIb + IIIc)} \tag{A8}$$

$$\#rings_{tetra} = \frac{(Ib + 2 * Ic)}{(Ia + Ib + Ic)} \tag{A9}$$

$$\#rings_{penta\ 5Me} = \frac{(IIb + 2 * IIc)}{(IIa + IIb + IIc)} \tag{A10}$$

$$\#rings_{penta\ 6Me} = \frac{(IIb' + 2 * IIc')}{(IIa' + IIb' + IIc')} \tag{A11}$$

$$DC = \frac{(Ib + 2 * Ic + IIb + IIb')}{(Ia + Ib + Ic + IIa + IIa' + IIb + IIb')} \tag{A12}$$

$$HP5 = \frac{IIIa}{(IIa + IIIa)} \tag{A13}$$

$$IBT = -log\left(\frac{IIa' + IIa'}{IIa + IIIa}\right) \tag{A14}$$

$$CI = \frac{Ia}{(Ia + IIa + IIIa)} \quad (= fIa_{Meth}) \tag{A15}$$

**Code and data availability**

Biomarker and associated metadata is provided in the Supplement and will be archived at the PANGAEA data repository. Code for generating and plotting set-specific FAs in R is provided in the Supplement. Additional code, data, and calibration equations will be available upon request.



**Author contribution**

GHM, JS, ÁG, JHR, SEC, DJH, and GdW designed the study and carried out the sampling. GHM, JS, ÁG, SK, and SEC funded the research. JHR, AB, and DJH performed the laboratory work and processed the HPLC-MS data under the supervision of JS. JHR and SK generated the R code. JHR analyzed the data and interpreted the results. JHR wrote the manuscript with input from all authors. All authors contributed to the article and approved the submitted version.

**Competing interests**

The authors declare that they have no conflict of interest.

**Acknowledgements**

This work was supported by the National Science Foundation (OPP-1737712 to GHM and JS; OPP-1836981to GHM, ÁG, and JS; DDRI-1657743 to GHM and SEC; EAR-1945484 to SK), a Doctoral Grant from the University of Iceland and a project grant from the University of Iceland Research Fund to ÁG, a National Geographic Society Early Career Grant (#CP-019ER-17 to SEC), and the University of Colorado Boulder. We thank the Inuit of Nunavut for permitting access to their land and to sample soils and lake sediment (Scientific Research Licenses 01022 17R-M, 02034 18R-M, 02038 19R-M) and the Qikiqtaani Inuit of Qikiqtarjuaq and Clyde River for assistance in the field. We thank the Nunavut Research Institute for logistical assistance and Polar Continental Shelf Project for air support. Field research in Iceland benefited from the assistance of Ian Holmen, Sveinbjörn Steinthorsson, and Thor Blöndahl. We thank Yuki Weber and Shucheng Xie for providing additional temperature data, James Russell for a valuable discussion, and Lina Pérez-Angel for assistance with the calibration code. We additionally thank Nadia Dildar, Sebastian Cantarero, and Katie Rempfert for laboratory assistance. The manuscript benefited from field assistance and/or discussions with Martha Raynolds, Shawnee Gowan, Helga Bueltmann, Elizabeth Thomas, Devon Gorby, Kayla Hollister, Kurt Lindberg, Nicolò Ardenghi, and Jamie McFarlin.

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

connections with the North Atlantic enhanced the deglacial warming in northeast China. Geology 45, 1031–
1034.
