# Peer review of "Revised fractional abundances and warm-season temperatures 1 substantially improve brGDGT calibrations in lake sediments 2"

_Biogeosciences, 2021_

## Referee Comment (RC1)

**General comments**

The manuscript of Raberg and colleagues reports the investigation of the impact of different environmental parameters on the distribution of branched GDGTs (brGDGTs) in a globally distributed set of lake sediment samples. Following this analysis, the authors propose new calibrations for the use of brGDGTs as paleo-proxies. Branched GDGTs are lipid biomarkers that are ubiquitous in continental settings and are increasingly used in paleo-studies to reconstruct past air temperatures (and sometimes pH) from lacustrine archives. However, the organisms producing brGDGTs are still unknown so the relationships between their distribution in a sample and environmental parameters (temperature, pH …) remain empirical. In this context, Raberg and colleagues provides a very comprehensive analysis of the relationships between the lipid distribution and a wide range of environmental parameters. Furthermore, they extend the latitudinal coverage of the worldwide sample set typically used to establish brGDGT calibrations to high latitude lakes. This study is thus of great interest for the community. The manuscript is clear and well written although it would benefit from reducing some parts (see below). I thus consider the manuscript to be suited for publication in *Biogeosciences* after minor revisions. The authors will find below a list of specific scientific comments and a list of technical ones.

**Specific comments**

-The approach of dividing the compounds into subsets to isolate each structural variation is interesting and valuable as it enabled the authors to reveal some physiological links between the lipid structures and the environmental parameters. I would suggest the authors to more specifically explain the rationale for their groupings in the introduction (l. 111-112). It would also be interesting to further discuss the relationships revealed by their approach in light with the literature on other biomarkers, such as isoprenoid GDGTs or alkenones. Are the observed lipid structure adaptations coherent with the homeoviscous membrane adaptation theory?

-Moreover, I strongly recommend the authors to better emphasize why the calibrations they set up are better than the previously established ones, especially the one, still under review but available as preprint, proposed by Martinez-Sosa and colleagues. I am, up to now, not convinced that the authors temperature calibration would perform better than others. Eventually, the paleo-community needs to know which calibration is the best suited for their archive(s). The authors should thus clearly and specifically state in their introduction and in the discussion part, the benefits for the paleo-community of using the calibrations they defined in the present manuscript over the others previously published calibrations.

-The proposed calibration with the conductivity of the lake water column is novel and of high potential for paleo-studies. The authors note that, in their dataset, pH and conductivity covary but they never suggest an explanation for this covariation and treat them separately all along the manuscript. In fact, it is not surprising to observe links between pH and conductivity in a lake water column. This aspect should be further discussed in the manuscript.

-In line with the previous comments, parts three and four seemed often redundant and sometimes too descriptive. The manuscript will benefit from a reorganization/condensation of these two parts. This reorganization should put forward comparisons of the study results with previous literature.

-In the introduction, the authors should briefly describe the four temperature indices they used and the differences between them (l. 114).

-Also, the authors should mention in the introduction the previous studies that evidenced the multiple sources of brGDGTs in lake sediments: from the lake catchment but also from *in situ* production in the water column or in the sediment. This aspect will have implications to define the environmental parameters the producers are effectively experiencing and could warrant further discussion in their discussion part notably in l. 592 (can the depth habitat of the producer have a role?) and in l. 601-614 (could the export mechanism also play a role here?).

-In the material and methods part, I wonder if the paragraph 2.3 really belongs there. Maybe the authors could put it in the supplementary material instead. Also, some details on their statistical procedure are missing. It would be important to know if their variables were all normally distributed or if the authors transformed and centered them before defining the linear models, for example.

**Technical comments**

l. 54: move "indices" before "methylation"

l. 63-64: "e.g." should be added before the reference cited

l. 67: "of the dependencies of brGDGTs on…"

l. 114: no S at "temperature"

l. 377: provide reference for the existing correlation.

Figure 7: replace r² by $R^2$ in the caption. In the legend *p-value* should be italicized and with a small p.

l. 428: $R^2$ is relevant only to evaluate the quality of a regression model. To discuss the correlation between two variables it is more appropriate to mention the correlation coefficient (r) and its *p-value*.

l. 459: explain what DC stands for

l. 593: same remark for HP5

---

## Author Response (AR1)

**Response to Reviewer 1 Comments (RC1):**

*We thank Reviewer 1 for their insightful comments and suggestions. A detailed response is provided below:*

Review of Raberg et al.,

**General comments**

The manuscript of Raberg and colleagues reports the investigation of the impact of different environmental parameters on the distribution of branched GDGTs (brGDGTs) in a globally distributed set of lake sediment samples. Following this analysis, the authors propose new calibrations for the use of brGDGTs as paleo-proxies. Branched GDGTs are lipid biomarkers that are ubiquitous in continental settings and are increasingly used in paleo-studies to reconstruct past air temperatures (and sometimes pH) from lacustrine archives. However, the organisms producing brGDGTs are still unknown so the relationships between their distribution in a sample and environmental parameters (temperature, pH ...) remain empirical. In this context, Raberg and colleagues provides a very comprehensive analysis of the relationships between the lipid distribution and a wide range of environmental parameters. Furthermore, they extend the latitudinal coverage of the worldwide sample set typically used to establish brGDGT calibrations to high latitude lakes. This study is thus of great interest for the community. The manuscript is clear and well written although it would benefit from reducing some parts (see below). I thus consider the manuscript to be suited for publication in *Biogeosciences* after minor revisions. The authors will find below a list of specific scientific comments and a list of technical ones.

**Specific comments**

-The approach of dividing the compounds into subsets to isolate each structural variation is interesting and valuable as it enabled the authors to reveal some physiological links between the lipid structures and the environmental parameters. I would suggest the authors to more specifically explain the rationale for their groupings in the introduction (l. 111-112). It would also be interesting to further discuss the relationships revealed by their approach in light with the literature on other biomarkers, such as isoprenoid GDGTs or alkenones. Are the observed lipid structure adaptations coherent with the homeoviscous membrane adaptation theory?

Original Response:

*We thank the reviewer for these suggestions. We will briefly outline our rationale behind the brGDGT groupings in the introduction, leaving the main discussion for Section 3. We will discuss connections to homeoviscous membrane adaptation theory for isoGDGTs and alkenones as a potential explanation for brGDGT structural distributions. However, we note that this connection is purely hypothetical until source organisms can be reliably cultured. A more involved discussion is beyond the scope of this work, but readers are directed to the Weijers et al. (2007) reference for a more detailed treatment of this subject.*

Addressed in revised manuscript:

*We have rephrased L124-127 to more clearly indicate the rationale behind the brGDGT groupings. We have added a brief discussion of the physiological implications of our results in the Conclusions section, but found a full discussion of homeoviscous membrane adaptation theory for brGDGTs, isoGDGTs, and alkenones to be beyond the scope of this work. Instead, we encourage future study.*

-Moreover, I strongly recommend the authors to better emphasize why the calibrations they set up are better than the previously established ones, especially the one, still under review but available as preprint, proposed by Martinez-Sosa and colleagues. I am, up to now, not convinced that the authors temperature calibration would perform better than others. Eventually, the paleo-community needs to know which calibration is the best suited for their archive(s). The authors should thus clearly and specifically state in their introduction and in the discussion part, the benefits for the paleo-community of using the calibrations they defined in the present manuscript over the others previously published calibrations.

Original Response:

*We thank the reviewer for this concern. We agree that "better" and "best" are ambiguous terms for evaluating our results that should be avoided. It is more accurate to say that our calibrations have some advantages in comparison to previous ones. In terms of $R^2$, the structural set calibrations perform comparably well to those derived using the standard approach (slightly worse for temperature and pH, slightly better for conductivity). The main advantage of using the structural set calibrations is that they can perform this well using a smaller number of compounds chosen to isolate variations in a single structural variable. In theory, this selective approach could provide some protection against the unwanted influences of other environmental parameters on different compounds (e.g. L618-621). Furthermore, subset-specific calibrations using only the most abundant compounds make it easier to apply brGDGT proxies to organic-lean samples, where only some (major) compounds are typically present above detection limit. However, we fully recognize the need to test these calibrations in paleoclimate archives, ideally with independent proxies for comparison. Such tests will be the subject of future work from our group, and we hope from the broader paleoclimate community as well. We have included the Full set temperature calibration (Eqn. 11) for this reason, but will emphasize this point in the text.*

Addressed in revised manuscript:
***We have checked for subjective references to calibrations performance and replaced any with explicit discussions of the advantages of the new method. These changes were made to highlighted sections in the Conclusions and the preceding paragraph.***

-The proposed calibration with the conductivity of the lake water column is novel and of high potential for paleo-studies. The authors note that, in their dataset, pH and conductivity covary but they never suggest an explanation for this covariation and treat them separately all along the manuscript. In fact, it is not surprising to observe links between pH and conductivity in a lake water column. This aspect should be further discussed in the manuscript.

Original Response:
*We thank the reviewer for this suggestion. Indeed, it is not surprising to find that conductivity and pH covary in our dataset. Variations in the concentration of $H_3O+$ (or $OH-$) will affect conductivity both directly and indirectly through altering the solubility of various ions. Furthermore, both pH and conductivity can be controlled by factors such as rock dominance and precipitation chemistry (Wetzel, 2001). While it is therefore not surprising that these variables covary in our dataset, the exact nature of their connection is complex and varies from one site to the next. An in-depth analysis of these connections is beyond the scope of this study; however, an understanding of modern pH and conductivity controls on a given site would be crucial for interpreting a downcore reconstruction of these variables using our proposed calibrations. We will therefore discuss the connection between pH and conductivity in the revised manuscript, both in the context of our modern dataset and for consideration during future downcore applications.*

*Wetzel, R.G., 2001. Salinity of Inland Waters, in: Wetzel, R.G. (Ed.), Limnology. Academic Press, pp. 169–186.*

Addressed in revised manuscript:
***We have introduced conductivity and pH and being fundamentally related, both in our dataset and in theory, at the start of Section 4.3. We have added a paragraph (starting at L665) that recommends examining both conductivity and pH trends in a downcore application, since they may vary in tandem, and understanding the site-specific relationship between the two variables in the modern day.***

-In line with the previous comments, parts three and four seemed often redundant and sometimes too descriptive. The manuscript will benefit from a reorganization/condensation of these two parts. This reorganization should put forward comparisons of the study results with previous literature.

Original Response:
*We thank the reviewer for this feedback. We recognize that the separation of Section 3 ("Partitioning brGDGTs into structural sets for FA calculations") from Section 4 ("Results and Discussion") results in some redundancies; however, we choose not to merge/reorganize these sections because they serve distinct purposes. Section 3 aims to outline the new structural set methodology, while Section 4 applies the methodology to the lake sediment*

*dataset. We will, however, work to condense and/or eliminate any redundancies in these sections where we are able. Comparisons of our results with previous literature are generally deferred to Section 4.4.4 ("Recommended Calibrations"), which aims to gather all applicable calibration results for clarity and convenience.*

Addressed in revised manuscript:
***No changes.***

-In the introduction, the authors should briefly describe the four temperature indices they used and the differences between them (l. 114).

Original Response:
*We thank the reviewer for this suggestion. We will add a brief description of the four temperature indices and their differences in the introduction, leaving a more detailed discussion for Section 2.4.*

Addressed in revised manuscript:
***While outlining all four temperature indices proved to disrupt the flow of the Introduction, we have provided a conceptual overview and justification for their use in L98-100. The detailed discussion is available in Section 2.4.***

-Also, the authors should mention in the introduction the previous studies that evidenced the multiple sources of brGDGTs in lake sediments: from the lake catchment but also from *in situ* production in the water column or in the sediment. This aspect will have implications to define the environmental parameters the producers are effectively experiencing and could warrant further discussion in their discussion part notably in l. 592 (can the depth habitat of the producer have a role?) and in l. 601-614 (could the export mechanism also play a role here?).

Original Response:
*We thank the reviewer for this suggestion and agree that a discussion of the various sources of brGDGTs to lake sediments would be valuable to include. We will do so in the revised manuscript.*

Addressed in revised manuscript:
***We have added a discussion of brGDGT sources to the Introduction (L111-121). We believe it sufficiently informs the discussion pointed out by the reviewer and have therefore not altered that text.***

-In the material and methods part, I wonder if the paragraph 2.3 really belongs there. Maybe the authors could put it in the supplementary material instead. Also, some details on their statistical procedure are missing. It would be important to know if their variables were all normally distributed or if the authors transformed and centered them before defining the linear models, for example.

Original Response:
*Paragraph 2.3 (Comparison of ASE and BD extraction methods) outlines a methodological result that is auxiliary to the main points of the paper, but we believe will be of interest to the brGDGT community. We opted to put it in the main text for this reason, but have no qualms moving it to the Supplement and will leave the final decision to the discretion of the editor. We note that the paper is within space constraints with the section included.*

*The dataset was constructed from our own data and the published studies available to us and were therefore not normally distributed. Logarithmic transformations for some variables (e.g. conductivity) resulted in a much more even spread of the data. BrGDGT data were normalized through fractional abundance calculations. No other transformations were performed.*

Addressed in revised manuscript:
***We opted to leave Section 2.3 in the main text. We added a note that the logarithmic transformations resulted in a more even spread of the data for some variables (L238).***

**Technical comments**

l. 54: move "indices" before "methylation"
l. 63-64: "e.g." should be added before the reference cited
l. 67: "of the dependencies of brGDGTs on…"
l. 114: no S at "temperature"
l. 377: provide reference for the existing correlation.
Figure 7: replace r2 by R2 in the caption. In the legend *p-value* should be italicized and with a small p.
l. 428: R2 is relevant only to evaluate the quality of a regression model. To discuss the correlation between two variables it is more appropriate to mention the correlation coefficient (r) and its *p-value*.
l. 459: explain what DC stands for l. 593: same remark for HP5

Original Response:
***We thank the reviewer for these technical comments. All will be addressed in the revised manuscript.***

Addressed in revised manuscript:
**We addressed all technical comments except for the comment on L114, for which we did not find an "S" at "temperature".**

**Reviewer 2 Comments (RC2):**
Raberg et al. investigate the relative abundance of branched GDGTs in a number of high- latitude lakes, and compare those distributions to previously published datasets from the tropics and mid-latitude sites. Through this work, they empirically derive global calibrations of the brGDGT distributions to temperature, salinity, and pH for use in paleoclimate reconstruction. The authors provide a comprehensive analysis of the existing and new data and develop a number of new indices to quantify the distributions of brGDGT abundances, some of which improve our understanding of how the lipid structures vary in response to environmental conditions. Overall, the authors have done a very good job and the new methods will be of wide interest to organic geochemists and the paleoclimate community. I recommend publication with minor but important revisions that I hope will improve the manuscript.

General notes:

1) The authors propose new ways to quantify brGDGT abundances through the use of 'sets' of brGDGTs with similar structures. This is a novel approach, although ultimately the authors seem to fall back on our existing understanding of how the different structures relate. Whatever the case, the authors should explain how and why they grouped the different sets. In addition, the authors examine a lot of different sets and report all of them, even though only some appear to be useful. The different ratios become a bit overwhelming by the end of the manuscript, and even though the authors should be commended for being comprehensive I'd suggest trying to limit the discussion to only the sets that ultimately proved useful.

Original Response:
***We thank the reviewer for this comment. The structural sets were formed solely to isolate each structural variable in turn. The approach was motivated by observations showing the relationship of methylation number with temperature and cyclization number with pH, but we decided to build a framework to systematically examine all structural variations in turn. We will clarify this in the text.***

***We understand that due to this systematic approach, we generate a large number of structural sets and associated nomenclature. However, a major goal of this publication is to outline the new structural set framework in full. While the Meth and Cyc-Isom sets appear to be the most important for this dataset, for example, subsequent research may find other sets to be more relevant. At this early stage of method development, we therefore opted to show and discuss the complete structural set framework.***

Addressed in revised manuscript:

*We have clarified how and why we grouped the different structural sets at the end of the introduction (L124-127). A more expanded explanation is already present in Section 3, including the opening paragraph and the topic sentences introducing each structural set.*

2) The authors suggest that the 'best' temperature calibration is based on a set of methylated brGDGTs (the "meth" set, maybe not the ideal name) to the average temperature of the months above freezing. It would be worth some more thinking and text about what "best" means. Ultimately, these calibrations will require extensive 'field testing' in different lacustrine environments to determine what works and what doesn't (just because a calibration 'works' with surface sediment does not mean it will produce meaningful downcore reconstructions). And what makes one calibration 'best? Although the calculations of fractional abundances within brGDGT "sets" are a new approach and are discussed at length in the paper, ultimately the best temperature regression (as measured by RMSE) uses the traditional method with all 15 brGDGTs to calculate the abundances. Lastly, although the 'months above freezing' is meaningful in regions with strong temperature seasonality it is not a particularly meaningful concept in low-latitude areas where there is no monthly variation; thus, the calibration target might not be 'best' in these regions. It would be worth it to consider these issues more in the conclusions.

Original Response:
*For the first portion of this comment, we refer the editor to Reviewer 1's similar comment and our reply, which is reproduced below:*

*"We thank the reviewer for this concern. We agree that "better" and "best" are ambiguous terms for evaluating our results that should be avoided. It is more accurate to say that our calibrations have some advantages in comparison to previous ones. In terms of $R^2$, the structural set calibrations perform comparably well to those derived using the standard approach (slightly worse for temperature and pH, slightly better for conductivity). The main advantage of using the structural set calibrations is that they can perform this well using a smaller number of compounds chosen to isolate variations in a single structural variable. In theory, this selective approach could provide some protection against the unwanted influences of other environmental parameters on different compounds (e.g. L618-621). Furthermore, subset-specific calibrations using only the most abundant compounds make it easier to apply brGDGT proxies to organic-lean samples, where only some (major) compounds are typically present above detection limit. However, we fully recognize the need to test these calibrations in paleoclimate archives, ideally with independent proxies for comparison. Such tests will be the subject of future work from our group, and we hope from the broader paleoclimate community as well. We have included the Full set temperature calibration (Eqn. 11) for this reason, but will emphasize this point in the text."*

*We agree that Months Above Freezing is only meaningful for cold sites, especially those with strong seasonality. At lower latitudes, where MAF is identical to MAT, the distinction is meaningless. In such cases, regional calibrations using lakes that do not go below freezing may indeed be more desirable. However, we note that our high-latitude sites from the Canadian Arctic overlap with high-altitude sites from East Africa in almost every MAF plot (e.g. Fig 7). This result suggests that using MAF instead of MAT allows data from high-latitude regions to be leveraged for calibrations that are relevant in the tropics. As noted in the text, temperatures reconstructed using the MAF calibrations might be interpretable as MAT in warm or low latitude regions. We therefore suggest using the MAF calibrations in paleoclimate applications and equating the resulting MAF temperatures with MATs where appropriate.*

Addressed in revised manuscript:
*We have addressed the first portion of this comment in response to Reviewer 1's similar comment. The second portion of the comment was already addressed in the text, as mentioned in our response, so we did not make additional changes.*

3) The Fit to MAF is interesting in that most sediment trap studies suggest brGDGT fluxes peak during events. Admittedly, "most" means only 2-3 studies, and it is not clear to me that those results can be reconciled with good fit to MAF. Does the improved fit to MAF imply that seasonal production and fluxes of brGDGTs are indeed biased toward summer, and if so, what are we missing from sediment trap studies? If calibrations are done to 'shoulder

season' temperatures, are the calibrations worse? The authors briefly discuss these issues, but it would be interesting to dig a bit deeper.

Original Response:
*We thank the reviewer for this comment and agree that much remains to be reconciled between sediment trap studies and calibration efforts. With sediment trap (and sub-annually resolved water filtrate) studies showing brGDGT production to vary temporally and to be tied to numerous potential influence, it is somewhat remarkable that the pool of brGDGTs in a lake surface sediment relates to an average air temperature at all. We, too, noted that MAF is probably roughly equal to shoulder-season temperature in most cases. Unfortunately, it proved difficult to standardize a "shoulder-season" across this varied dataset. We tried calibrating against the first and last positive-degree months, or the first and last two, but our monthly temperature resolution proved to be too coarse. Additionally, if shoulder-season temperatures are relevant, it is likely due to overturning events and/or seasonal nutrient fluxes occurring at these times, which will vary widely between lakes. We thus abandoned these efforts. However, we will expand our discussion to include this subject in the text.*

Addressed in revised manuscript:
*We added a mention of our attempts to quantify shoulder season temperatures at the end of Section 2.4.*

4) I have some concerns about the section starting on line 580 that develops calibrations for dissolved oxygen. Although this is inconclusive, I question whether we the environmental data is good enough to meaningfully address this. Is DOmean the average of the entire water column? And over what seasons? Calculating this variable is a fraught exercise without access to each lakes' actual DO profiles, many of which do not appear to exist.

Original Response:
*We agree with the reviewer's concerns, and echo them in the discussion at the end of the section. Given the increasing evidence of a dissolved oxygen influence, we believe the section on DO will be of interest to readers and will highlight the need for better DO data. However, given that the results are inconclusive, we will discuss the possibility of moving the technical aspects of this discussion to the Supplement with the editor.*

Addressed in revised manuscript:
*We have moved the technical aspects of the section on DO to the supplement (section S5.1).*

5) Several of the indices and calibrations utilize fractional abundances of the bicyclized penta- and/or hexamethylated compounds that are often not abundant in sufficient quantities to be accurately measured in many of the published datasets. It is not clear how (a) the authors treated those sites in their calibrations, (b) what their limits of detection are for the different compounds in their new sites and how those were determined, and (c) what they suggest to do at sites where these compounds (chiefly IIc, IIc', IIIb, IIIb', IIIc, IIIc') are below detection levels. This could create problems for some of the sets that isolate these structural groups.

Original Response:
*We thank the reviewer for this important concern and note that it is relevant to both traditional calibrations and those constructed using the new structural sets. For applications involving samples with compounds below the detection limit, we believe that our new method actually provides a unique advantage by allowing for the construction of calibrations that rely solely on high-abundance compounds (e.g. Ia, Ib, Ic, IIa, IIa', IIIa, IIIa'; Eqns. S1, S4, and S7).*

*We treated compounds below the detection limit as having an absolute abundance of zero. This assumption led some sites to have compounds with FA = 0 and/or FA = 1. All of these FAs are plotted as such (e.g. Fig. 7c) and tend to be associated with noisier trends (presumably due in part to the lower abundances). Removing these sites from the dataset would have removed valuable points from the stronger trends (e.g. Fig. 7a), upon which our calibrations primarily rely. For subset-specific calibrations (e.g. Eqns. S1-S9), we did not face this issue and removed samples with any FA = 1.*

*We will add the above discussion, along with a description of our own detection limit, to the Methods section. We will also stress the potential applications for low-abundance samples in the Discussion section.*

Addressed in revised manuscript:
*We have added a description of our detection limit to the section 2.2 (L177). We have alerted readers to potential pitfalls of using our main calibration (Eq. 10) in organic-lean samples and offered alternatives via the subset-specific calibrations. We also added a new subset-specific calibration (Eq. S1) that uses both non- and monocyclized compounds, as bicyclized compounds are the most common to fall below the detection limit.*

6) Line 432. The text below highlights a challenge in working with these data – the environmental variables themselves are strongly correlated. This makes it difficult to conclusively support some of the statements in this paragraph – "temperature may therefore play a secondary role"... for instance. This assumes that temperature actually does play a role, rather than that temperature is correlated to pH and conductivity, and therefore is (spuriously) correlated to the brGDGTs. It might be worth considering application of multivariate statistical methods that take into account collinearity among the predictors, such as redundancy analysis, to address this issue.

Original Response:
*We agree with the reviewer – correlations between environmental variables pose an inherent challenge to this type of work. We will add a correlation table for the environmental variables in the Supplement and expand our discussion in this section and elsewhere in the text. We further agree that redundancy analysis and other multivariate statistical methods would be a valuable extension of this work and will be the subject of a future publication.*

Addressed in revised manuscript:
*We have added a correlation table for the environmental variables (Table S2). We have revised the lines pointed out by the Reviewer.*

Minor things:

Line 59. Why "however"? The shifts in community composition can track shifts in environmental conditions.

Original Response:
*We use "however" to indicate that changes in community composition could affect brGDGT distributions independently of a physiological response.*

Addressed in revised manuscript:
*No changes.*

Line 74. As stated above, note that, at many sites, the highly cyclized brGDGTs are often below detection levels, such that equation 2 is reduced to the fractional abundances of only the most abundant compounds.

Original Response:
*We will include a discussion of this point either here or later in the text.*

Addressed in revised manuscript:
*A discussion of compounds below the detection limit was added in response to the reviewer's earlier comment.*

Line 138-139. How many sites had water chemistry data?

Original Response:

*We will include this information.*

Addressed in revised manuscript:
*We have added a table with this information in the supplement (Table S1).*

Line 220. Did the logarithmic scaling result in a normal distribution of these environmental variables?

Original Response:
*Logarithmic scaling greatly improved the normality of these variables, though they still did not pass the Shapiro-Wilks normality test. We will include this information.*

Addressed in revised manuscript:
*We have included this information in L238.*

Line 301. Here and elsewhere throughout the text, it would be best to provide p values for the correlations you report.

Original Response:
*Due to the large number of $R^2$ values discussed in the text, we found that including accompanying p-values made the text overwhelming and more difficult to follow. As the vast majority of p-values were < 0.01, we opted to only report p-values ≥ 0.01. This explanation is included in the Methods.*

Addressed in revised manuscript:
*No change.*

Line 398. There is some thinking that a substitution closer to the head group would be more influential on membrane fluidity (structural disruption closer to the surface of the compound). Your results suggest this is a minor effect.

Original Response:
*We thank the reviewer for this interesting comment. We would appreciate if the reviewer could provide some references to us so we could explore this topic in our dataset.*

Addressed in revised manuscript:
*We have added a discussion of this topic in Section S3 in the Supplement and briefly referenced it in the text (L416-17).*

Line 427. This does not suggest they have similar influences on brGDGTs. Rather, it suggests that conductivity and temperature covary in the environment.

Original Response:
*We thank the reviewer for pointing this out. We agree and will change this text.*

Addressed in revised manuscript:
*Changed.*

Line 433. Could you cite a table or similar for this statement?

Original Response:
*We will add a reference to the supplemental figures containing the relevant statistical values.*

Addressed in revised manuscript:
*We added a reference to Figures 8 and S23.*

Line 471. Similarly, how did you assess that ring number is "primarily" correlated with pH rather than conductivity? Based on an r2 difference of 0.04?

Original Response:
*We agree that an $R^2$ difference of 0.04 is small. However, this (or a comparable) difference was apparent across almost all 15 compounds (Fig. 9 vs. Fig. S20).*

Addressed in revised manuscript:
*No change.*

Line 499. Regressions of the temperature variables onto what? Which brGDGT formulations?

Original Response:
*We will change the sentence beginning on L499 to read: "For each structural set and subset defined in Section 3, we performed regressions of subset-specific brGDGT FAs against five temperature variables to generate multiple global-scale calibrations."*

Addressed in revised manuscript:
*We implemented the change described.*

Line 505. A challenge here is that MAT = MAF in the low latitudes. The main reason for the MAF formulation is to account for the additional complexity introduced by a global calibration that includes high latitude systems. It would be worthwhile to reemphasize this here.

Original Response:
*We will reemphasize this nuance in the text.*

Addressed in revised manuscript:
*We reemphasized this nuance in L527-529.*

Line 540. This would make some sense, since MAT = MAF in the sites included in the Russell et al. (2018) calibrations.

Original Response:
*Agreed.*

Addressed in revised manuscript:
*No change.*

Line 541. "On a regional scale"... It would be worthwhile to mention the relatively unique (highly alkaline) nature of these lakes.

Original Response:
*We thank the reviewer for noting this and will include it in the text.*

Addressed in revised manuscript:
*We have included a mention of the high alkalinity of these lakes in this sentence.*

Figure 10. It is a bit difficult to figure out what is being plotted in these figures, in particular panel a. It would be most helpful if the equations are introduced alongside this figure. Related to this, it appears from the text that a very simple formulation, MBT'5Me, performs nearly as well as more mathematically complicated (e.g. equation 10)

versions. Is this correct? Lastly, although a high r2 is great, the RMSE is in some ways more important to climate reconstruction. It might be good to include a plot showing the RMSE, similar to figure 10A.

Original Response:
*We thank the reviewer for these suggestions. We will explore the option of introducing the equations alongside the figures.*

*It is true that MBT'$_{5Me}$ performs nearly as well as Eq. 10. This is perhaps not surprising, as the Meth FAs in Eq. 10 were inspired by the same underlying principle as the MBT'$_{5Me}$ index – methylation number varies in response to temperature. We do not suggest abandoning the MBT'$_{5Me}$ index, but we present the Meth structural set as a more targeted approach at isolating the underlying connection between methylation number and temperature.*

*We thank the reviewer for pointing out that a plot similar to Fig. 10A that compares the fits using RMSE rather than R2 will be of interest to the paleoclimate community. We will explore options for including such a plot, either as part of the main text figures (Fig. 10 and 11) or in the Supplement.*

Addressed in revised manuscript:
*We decided to keep the structure of the manuscript as is. We have included a plot similar to Fig. 10A that compares the fits using RMSE rather than R$^2$ in the Supplement (Fig. S41).*

Line 674. This is correct in a way, but the use of structural sets does not ultimately improve calibration performance. Correct?

Original Response:
*We will clarify that the structural sets "improve compound-specific correlations with environmental parameters..." in line 674. It is true that they perform comparably well to traditional methods in terms of R$^2$. However, we believe the fact that the structural sets can perform this well while isolating only the relevant structural changes is an "improvement". This point is currently ambiguous in the text, however. We thank the reviewer for pointing this out and will revise accordingly.*

Addressed in revised manuscript:
*We have included the phrase in that line as described in our response. We have specified our meaning of "improvement" throughout the text (see previous reviewer comments).*

Line 692, could add a clause of "particularly in high-latitude environments".

Original Response:
*We will incorporate this helpful suggestion.*

Addressed in revised manuscript:
*We have incorporated this suggestion.*

The paper refers to papers by Sosa et al. (2020, 2020a) that are not in the references. Please provide the correct citations.

Original Response:
*The references are under Martínez-Sosa et al., rather than Sosa et al. However, we found references to both "Martínez-Sosa et al. (2020)" and "Martínez-Sosa et al. (2020b)" in the text, though we only intended to cite one publication by this author (the preprint, as submitted to Geochimica et Cosmochimica Acta). We will correct this.*

Addressed in revised manuscript:
*We have fixed this issue.*

**Editor Comments 1 (EC1):**

Additional comments from Associate Editor:

Thank you for submitting this very interesting manuscript to Biogeosciences. I think the reviewers have some good suggestions and raise interesting questions, some of which I also have, so won't repeat them here in detail.

Further, I wonder if you could provide more details about in situ production of brGDGTs in the lakes vs what is the imprint of brGDGT supply from the lake catchments. The origin of the brGDGTs will determine which parameters can be reconstructed, such as pH in soils vs in the water column, temperatures of different water depths, etc? I also think a more detailed discussion about the reasons for the apparent covariance between conductivity and pH in your dataset would be useful.

Original Response:

*We thank the editor for these comments and suggestions. We agree that the relative abundance of in situ lacustrine vs. catchment-derived brGDGTs has important implications for calibration studies and downcore applications. Much research has been invested to tease apart the sources of brGDGTs to lake sediments as a result. Unfortunately, even basic studies of the sources of brGDGTs to the lake surface sediment are missing for the vast majority of the lakes in our study, limiting our ability to address these issues here. This unavoidable source uncertainty likely enters our study as scatter in the data – a lake with significant soil input might be shifted from its "pure lacustrine" position. We will include a discussion of this topic in the text. We will also discuss the covariance between conductivity and pH, as outlined in our response to Reviewer 1's comment.*

Addressed in revised manuscript:

***We have included a discussion of brGDGT sources in the introduction, per your and a previous reviewer's comment (L111-121).***

I also wonder if there should be more information about global vs regional/location calibrations for the reader that is less familiar with the topic and to explain why you rather extend the global calibration instead of developing another regional calibration that better capture the strong seasonality.

Original Response:

*We thank the editor for this suggestion and agree that it will be a valuable point to discuss in the text. We initially set out to construct regional calibrations for our Canadian and Icelandic datasets. However, we found that brGDGTs were not able to adequately resolve temperature differences within the small MAF range (6.6°C, or 3.2°C without the warm and cold end-members) of these datasets. This is an important result as it indicates that other environmental and/or site-specific effects complicate the link between lake sediment brGDGT distributions and air temperature. We will discuss this in the text.*

Addressed in revised manuscript:
***We have added a discussion of regional calibrations to the end of section 4.4.1.***